# Changes in quality of life (QoL) and other patient-reported outcome measures (PROMs) in living-donor and deceased-donor kidney transplant recipients and those awaiting transplantation in the UK ATTOM programme: a longitudinal cohort questionnaire survey with additional qualitative interviews

Andrea Gibbons [1,2] Janet Bayfield,[2,3] Marco Cinnirella,[4] Heather Draper,[5] Rachel J Johnson,[6] Gabriel C Oniscu,[7] Rommel Ravanan,[8] Charles Tomson,[9] Paul Roderick,[10] Wendy Metcalfe,[7] John L R Forsythe,[7,11] Christopher Dudley,[8] Christopher J E Watson [12,13] J Andrew Bradley,[12,13] Clare Bradley [2,3]

► Prepublication history and additional materials for this paper is available online. To view these files, please visit the journal online (http://dx.doi.org/10.1136/bmjopen-2020-047263).

For numbered affiliations see end of article.

**Correspondence to**
Professor Clare Bradley;
cb@healthpsychologyresearch.com

## ABSTRACT

**Objective** To examine quality of life (QoL) and other patient-reported outcome measures (PROMs) in kidney transplant recipients and those awaiting transplantation.
**Design** Longitudinal cohort questionnaire surveys and qualitative semi-structured interviews using thematic analysis with a pragmatic approach.
**Setting** Completion of generic and disease-specific PROMs at two time points, and telephone interviews with participants UK-wide.
**Participants** 101 incident deceased-donor (DD) and 94 incident living-donor (LD) kidney transplant recipients, together with 165 patients on the waiting list (WL) from 18 UK centres recruited to the Access to Transplantation and Transplant Outcome Measures (ATTOM) programme completed PROMs at recruitment (November 2011 to March 2013) and 1 year follow-up. Forty-one of the 165 patients on the WL received a DD transplant and 26 received a LD transplant during the study period, completing PROMs initially as patients on the WL, and again 1 year post-transplant. A subsample of 10 LD and 10 DD recipients participated in qualitative semi-structured interviews.
**Results** LD recipients were younger, had more educational qualifications and more often received a transplant before dialysis. Controlling for these and other factors, cross-sectional analyses at 12 months post-transplant suggested better QoL, renal-dependent QoL and treatment satisfaction for LD than DD recipients. Patients on the WL reported worse outcomes compared with both transplant groups. However, longitudinal analyses (controlling for pre-transplant differences) showed that LD and DD recipients reported similarly improved health status and renal-dependent QoL (p<0.01) pre-transplant to

post-transplant. Patients on the WL had worsened health status but no change in QoL. Qualitative analyses revealed transplant recipients' expectations influenced their recovery and satisfaction with transplant.
**Conclusions** While cross-sectional analyses suggested LD kidney transplantation leads to better QoL and treatment satisfaction, longitudinal assessment showed similar QoL improvements in PROMs for both transplant groups, with better outcomes than for those still waitlisted. Regardless of transplant type, clinicians need to be

## Strengths and limitations of this study

► We examined a number of patient-reported outcomes in people requiring kidney transplantation, including quality of life, well-being, treatment satisfaction and health status.
► We collected pre-transplant data for a subsample of deceased-donor and living-donor kidney transplant recipients while they were still waiting for a transplant, and assessed outcomes from pre-transplant to 1 year post-transplant.
► Controlling for various medical and demographic factors, including age, time on dialysis and education, impacts the findings, removing apparent benefits of living-donor kidneys over deceased-donor kidneys.
► The sample of kidney transplant recipients with pre-transplant data is relatively small (n=67 including 26 with living donors and 41 with deceased donors).
► Longer follow-up may be required to examine fully any differences in PROMs between groups post-transplant.

aware that managing expectations is important for facilitating patients' adjustment post-transplant.

## INTRODUCTION

Healthcare usually aims to improve health and minimise disability,[1] and there is growing acknowledgement that patient-reported outcome measures (PROMs) are important for evaluating the effectiveness of healthcare. PROMs are measures of outcomes that are directly reported by the patient, and are usually questionnaires that can be generic or condition-specific. PROMs may measure a wide range of outcomes including health status, quality of life (QoL), treatment satisfaction, wellbeing and other outcomes such as symptoms. Despite their importance in assessing the patient's perspective, PROMs such as QoL are not measured routinely as part of the evaluation of surgical interventions.[1] Part of the reason for this has been the belief that brief health-status tools (such as the EuroQoL-5D (EQ-5D) capture QoL,[1] as well as the time and cost involved in collecting and interpreting the data.[2 3] However, PROMs other than health status tools have an important role in clinical care[3] as they may determine the comparative benefits of different medical interventions used to treat the same condition.[4]

One example of a condition that can benefit from the measurement of PROMs is advanced chronic kidney disease (CKD Grade (G) 5). Kidney transplantation is commonly considered to be the best medical treatment for most people with advanced CKD G5 who are fit enough for the procedure.[5] Recent changes to the kidney donor system, such as deemed consent legislation in Wales since 2015, and a similar system introduced in England in May 2020, and due to be enacted in Scotland in 2021, aim to increase the number of deceased-donor (DD) kidneys available.[6] Kidney transplant waiting times have also reduced in recent years due to an increase in unrelated living donation including altruistic donation, and kidney sharing schemes that allow for greater access to living-donor (LD) transplantation for blood-group or human-leucocyte-antigen-incompatible donor-recipient pairs.[7] Transplantation success is most often measured by patient and graft survival and as such LD transplantation provides a greater survival advantage when compared directly with DD transplantation, controlling for differences in variables such as age[8] and time on dialysis.[9] LD transplantation is associated with lower rejection rates[7] and spares individuals the uncertain wait for a deceased-donor organ.[7] However, it is important to note that LD recipients tend to have better clinical indices of health pre-transplant.[10 11] These potential confounding factors can impact graft survival,[12] so pre-transplant differences between LD and DD recipients should be measured and controlled, before comparisons are made between the groups. The other issue apparent with these studies is that they focus on the comparative benefits of LD and DD transplantation for health, but not for other outcomes such as QoL.

When PROMs have been assessed in kidney transplantation, cross-sectional post-transplant comparisons indicated that LD recipients reported greater social involvement and happiness compared with DD recipients.[12] When cross-sectional data are analysed, it is important to have well-matched controls but in practice that is difficult to achieve because those who are left on the waiting list for deceased-donor kidneys tend to have more health problems than those who receive a transplant. Studies that have controlled for underlying differences across the groups, such as differences in age or comorbid disease, report similar outcomes for both transplant groups on health status measures such as the SF-36,[13] although many LD recipients report experiencing feelings of guilt about the risks to their donor.[14 15] Longitudinal studies, which can include and control for baseline measures of outcomes before participants receive a transplant, suggest that improvements in health status can be seen in the first few months post-transplant for both DD and LD, but that these improvements remain stable after this time.[16–19] However, these studies lack any group with which to compare transplant recipients, such as those still awaiting transplantation, and not all include data pre-transplant. Those who are transplanted are more likely to be in better health than those patients still wait-listed for a transplant, so it is important to control for baseline differences between participants who receive a transplant and those who do not.

One major problem with all of these studies is that the instruments used to measure QoL such as the SF-36, actually measure health status.[20] Health status includes aspects of a person's life such as their physical ability, daily functioning and experience of symptoms. Health-related QoL is a term that is commonly and misleadingly used to refer to health status tools such as the SF-36 and EQ-5D, which may assess health and functioning but do not assess the impact of health and functioning on QoL. In contrast, QoL as measured by the renal-dependent QoL (RDQoL) measure used in Access to Transplantation and Transplant Outcome Measures (ATTOM),[21] is defined within the questionnaire as 'how good or bad you feel your life to be'. RDQoL, modelled on the Audit of Diabetes Dependent Quality of Life (ADDQoL) for diabetes[22] and influenced by McGee *et al's*[23] individualised interview method of generic QoL measurement, establishes the relevance of different aspects of life for each individual respondent and excludes non-relevant items from the scoring. It allows for individual differences in the importance of relevant aspects of life for QoL as well as measuring the impact of the renal condition on each relevant aspect of life. Impact is weighted by importance and an average weighted impact (AWI) score obtained for each individual indicating the nature and extent of the impact of the renal condition on the individual's QoL. A single item to measure generic QoL is also included in the measure and this item can be expected to be more strongly related to health status generally than the condition-specific AWI score.[24] Given the confusion

surrounding the definitions and measurement of these concepts, it is important that research includes genuine measures of QoL and condition-dependent QoL as well as a health status measure when investigating PROMs in people with CKD. The addition of qualitative research methods alongside PROMs, can be valuable in providing more detailed insight into how transplant recipients experience transplantation and how it impacts QoL.

The main objectives of the present study were (1) to examine QoL (generic and condition-dependent), health status and other PROMs in recipients of DD or LD transplants, and (2) to compare them over time with those on the waiting list (WL) for a DD transplant in a matched cross-sectional cohort and in a subsample with longitudinal data pre-transplant and post-transplant. Specifically, it was hypothesised that, controlling for potential confounding factors, there would be very few differences in outcomes between DD and LD recipients. It was also hypothesised that transplant recipients would report less negatively impacted QoL and better scores on other PROMs than participants waiting for a transplant. The third objective was to use qualitative interview methods to explore in more depth the expectations and experience of transplantation and how it influences QoL 1 year post-transplant in DD and LD transplant recipients.

## METHODS

### Study design, participants and procedure

This study employed a longitudinal cohort survey design, as well as a qualitative interview approach. The study was part of the UK ATTOM research programme.[25] ATTOM was based on a large national observational cohort study involving all kidney units in the UK in order to examine the reasons for disparities in access to renal transplantation.[26 27] Full ATTOM methodology has been reported elsewhere.[25] The current study was conducted within a work-stream examining detailed PROMs in patients fluent in English, aged less than 75 years, receiving renal-replacement therapy (RRT) who completed measures of health status, well-being, QoL and treatment satisfaction. Qualitative interviews were conducted with a subset of patients 1 year post-transplant.

Following ethical approval (East of England Research Ethics Committee 11/EE/0120), and informed consent, participants were recruited by ATTOM research nurses to the study as incident recipients of DD transplants (n=104), incident LD transplant recipients (n=94) or patients on the WL for a transplant (n=165) and followed up either 1 year post-recruitment or 1 year post-transplant (figure 1). Across 18 UK renal centres, a subset of participants recruited to ATTOM were recruited to this work-stream. The first eligible patient for each transplant group seen each month (November 2011 to March 2013) by each nurse was invited to take part. Inclusion criteria were fluency in English and <75 years of age. Participants were recruited if they had G5 CKD and received a DD or LD transplant within 1 month. The UK

Transplant Registry identified possible matched controls for recruited transplant recipients every 2 weeks, and members of this list of potential participants were then invited to take part as patients on the WL. Participants on the WL were matched to DD transplant recipients as closely as possible in terms of renal centre, age (±5 years), time on the WL (±100 days) and previous type of RRT. Participants completed measures of health status and well-being when first recruited to ATTOM by the research nurses, and measures of generic and renal-dependent QoL, and renal treatment satisfaction 3 months post-transplant/post-recruitment via telephone or post. A timeframe of 3 months was chosen to allow participants time to experience their treatment (ie, transplant for those receiving a transplant) and be able to reflect on how their QoL and treatment satisfaction were impacted after their return home for those having surgery. At 1 year, participants were contacted via telephone and completed all measures again (via telephone or post), as well as a measure of change in treatment satisfaction which compared satisfaction with current renal treatment and previous renal treatment. One-year follow-up was chosen because it would be expected that clinical outcomes would be stable 1 year post-transplant, but it is not clear whether non-clinical outcomes (ie, PROMs) would also remain stable. At the same time, choosing 1 year follow-up allows for comparisons to be made with previous research that has examined health status using the same timeline. A sample of 100 participants in each of the three groups was calculated as adequate to reach 80% power within the planned analyses. Some of the participants recruited as patients on the WL subsequently received a DD (n=41) or LD kidney transplant (n=26) in the year of follow-up. For these participants, the initial measures were completed while they were still on the WL, giving a pre-transplant measure, and the second set of measures was completed 1 year post-transplant (figure 1). These data allowed for secondary analyses in patients for whom we had true baseline measures comparing the two transplant groups and those remaining on the waiting list pre-transplant and post-transplant. Participants recruited as WL comparison patients who went on to receive a transplant within ATTOM were not asked to complete the measures at 3 months post-transplant in the same way as those recruited to the study as transplant recipients were asked to do. It was not always possible to contact people within 3 months of receiving their transplant once they had been recruited as patients on the WL, and a limit of 4 points of data collection were set for each participant, to minimise participant burden.

A subset of transplant recipients was selected for qualitative telephone interviews. The aim of this qualitative study was to explore the experience of receiving a transplant, and how having a transplant affects QoL and other PROMs. The sample consisted of 10 DD transplant and 10 LD transplant recipients. All participants had been contacted previously by authors (AG or JB) when arranging completion of questionnaires. Participants

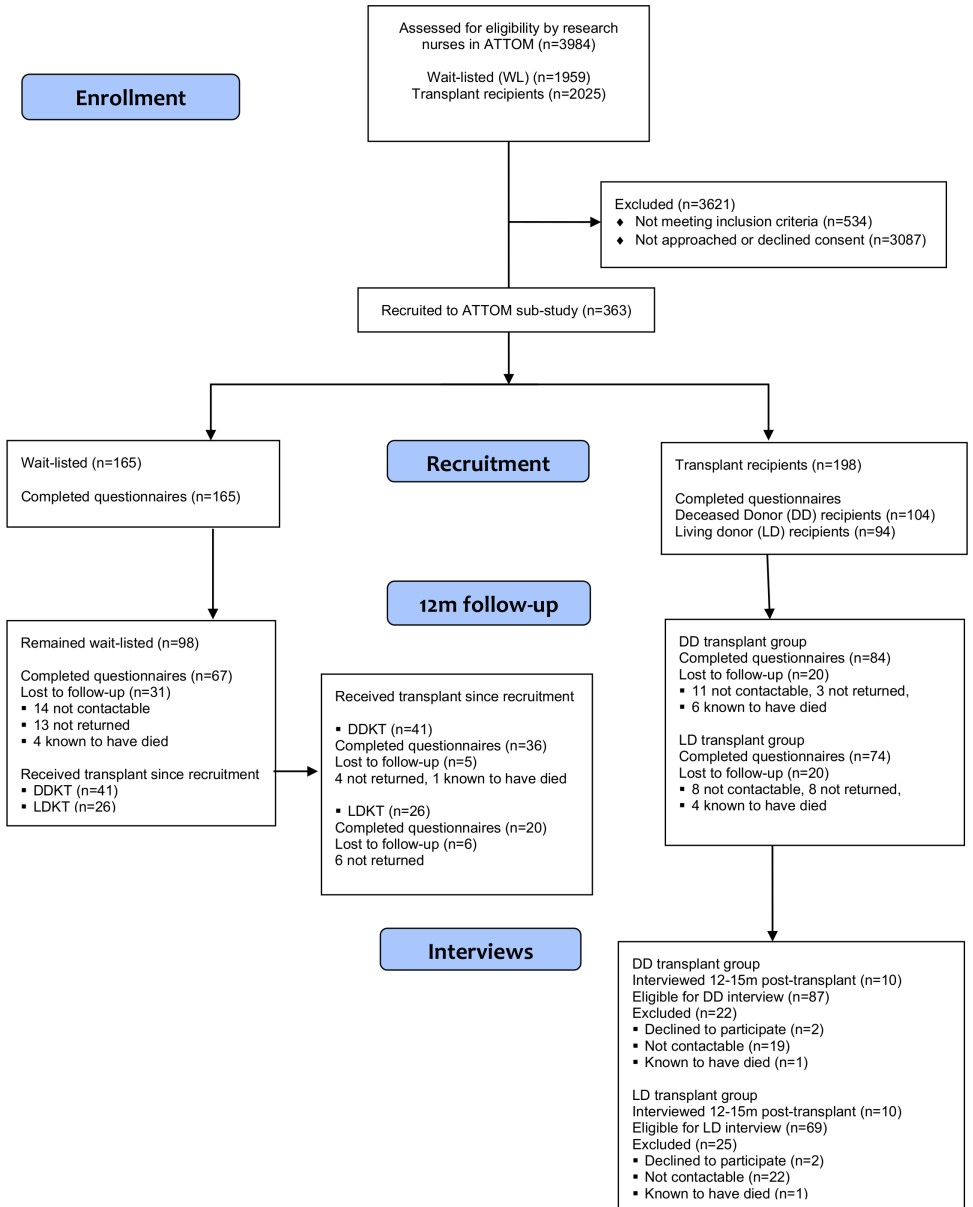

**Figure 1** CONSORT (Consolidated Standards of Reporting Trials) diagram.
ATTOM, Access to Transplantation and Transplant Outcome Measures; DDKT, DD kidney transplantation; LDKT, LD kidney transplantation; m, month.

were invited to take part based on their AWI scores on the RDQoL questionnaire,[21] completed 1 year post-transplant. Participants with AWI scores above or below one SD of the mean score, as well as those with a mean score, were included so that participants had varying levels of QoL. The qualitative sample was selected on the basis of QoL scores to aid our understanding of the range of responses and how they may relate to variations in QoL. The profile of the qualitative-study participants reflected the age and sex profile of the larger sample. Participants were informed that the interview would explore their questionnaire responses related to their QoL and treatment satisfaction. Semi-structured telephone interviews were conducted between January and August 2014, and were conducted by a single interviewer using an interview schedule (online supplemental data 1) guided by

published literature and participants' questionnaire responses.[28] Interviews were conducted by a postdoctoral research fellow (AG) experienced in qualitative research and trained in use of NVivo software (QSR International, USA) for qualitative analysis, with background in the field of health psychology. Field notes were made following every audio-recorded interview. Interview recordings (mean=52 min, SD=13.4, range=31–76) were transcribed.

**Patient and public involvement**

Patients were involved in the development of the ATTOM project objectives and procedures, as well as analysis and dissemination. A patient representative from the National Kidney Federation was involved in all aspects of the development of the research. A patient representative was also invited to all steering group meetings throughout

the duration of the project. Findings were presented to patients and lay people as part of an NIHR (National Institute for Health Research) stakeholder ATTOM meeting in November 2017.

## Outcomes

A summary of the primary outcome measures can be seen in online supplemental data 2, table 1. QoL was assessed using the RDQoL questionnaire,[21] developed from the ADDQoL questionnaire.[22 29] It includes one overview item to measure generic QoL on a scale from excellent (+3) to extremely bad (−3). Twenty-one items measure the impact of the renal condition on specific aspects of life. Each specific item includes a rating of the impact of the renal condition on QoL, from −3 (most negative impact) to +1 (positive impact), and a rating of importance of that aspect of life for the individual's QoL ranging from very important (+3) to not at all important (0). Impact and importance ratings are multiplied to give a weighted impact score for each item, ranging from −9 (most negative/important impact of the renal condition on QoL) to +3 (most positive/important impact). Some items include preliminary questions to determine applicability to the individual (eg, work). The weighted impact scores are summed and divided by the number of applicable items to give an AWI score, also ranging from −9 to +3. Principal axis factoring and reliability analyses confirmed that the RDQoL was appropriate for both transplant and non-transplant groups (see online supplemental data 3, tables 2–4).

Well-being was measured by the Well-being Questionnaire (W-BQ12).[30] Higher scores indicate greater well-being (range=36-0). Health status was measured by the EQ-5D-5L,[31] which has two sections. The first section includes five dimensions of health that are rated on five levels. These data were then converted into a population preference score called a health-utility value, using the new value set for England.[32] The second section asks participants to rate 'Your health today' on a visual analogue scale (EQ-VAS) from 100 (best health you can imagine) to 0 (worst health you can imagine). The EQ-VAS is a PROM, but the utility measure is not, as it depends not only on patients' reports but also on the values assigned by the general population to each profile of scores for the five dimensions of health used to create the value set for England.

Satisfaction with current renal treatment was measured using the Renal Treatment Satisfaction Questionnaire status version (RTSQs).[33] It has 13 items, with 7-point scales (6 to 0), that are summed to give a total score (range=78–0). Higher scores indicate greater satisfaction with current renal treatment. The change version (RTSQc) is modelled on the change version of the Diabetes Treatment Satisfaction Questionnaire (DTSQc),[34] developed to counteract ceiling effects found with the status version (DTSQs).[35] It asks participants to compare satisfaction with current treatment with satisfaction with their previous treatment on 7-point scales (+3 to −3 where 0=no

change) and is summed to give a total score from +39 (maximum satisfaction improvement) to −39 (maximum deterioration in satisfaction). Principal axis factoring and reliability analyses confirmed that the RTSQs and RTSQc were appropriate for both transplant and non-transplant groups (see online supplemental data 3, tables 5–6).

For access to the RDQoL, ADDQoL, RTSQs and c, DTSQs and c, and Well-being Questionnaire, visit https://www.healthpsychologyresearch.com/. For access to EQ-5D visit https://euroqol.org/.

All sociodemographic and medical information was recorded by research nurses from medical notes at the time of recruitment. Any condition mentioned in patients' records was identified and classified into groups (heart disease, heart failure, liver disease, diabetes, mental health problems, peripheral vascular disease). Prescribed immunosuppressant and steroid medication were also recorded. Height and weight were recorded to allow calculation of body mass index (BMI). Any subsequent changes to information (such as change from being on the WL to receiving a transplant) were self-reported by participants.

## Analysis

Reflecting previous research in this area, the main analyses involved cross-sectional differences at 1 year follow-up in those who received a DD transplant (n=145), LD transplant (n=120) and those who remained waiting for a transplant (n=98) at 1 year follow-up. These analyses were conducted using one-way analysis of covariance (ANCOVA), controlling for medical and demographic differences recorded at recruitment. Clinical factors such as previous renal replacement therapy, as well as demographic factors including age, and indicators of socioeconomic status (such as car ownership and qualifications), can influence a person's perspective on their QoL and treatment satisfaction. We therefore assessed these variables and controlled for them in the analyses if differences between groups were identified (using $\chi^2$ tests or analysis of variance).[36] It was hypothesised that patients still on the WL would have more negative scores on PROMs than those who received a transplant.

Secondary analyses involved a prospective assessment of those participants recruited as patients on the WL (n=165). The sample was divided into three groups based on their treatment at follow-up; those who remained waiting for a transplant (n=98); those who subsequently had a DD (n=41); and those who subsequently had an LD transplant (n=26). This allowed for changes in outcomes to be measured pre-transplant and post-transplant for these groups and for pre-transplant differences to be controlled for. Differences between groups and over time were investigated with a series of 3 (group) x 2 (time) ANCOVAs with planned comparisons on all PROMs and utility outcome scores (controlling for various factors). It was hypothesised that those who received a transplant would report improved PROMs, while those remaining on the waiting list would report either no change, or a

deterioration in scores. It was also hypothesised that when differences in sociodemographic factors are controlled, there will be no differences in outcomes between DD and LD recipients. A one-way ANCOVA was also conducted to examine differences between these three groups (those who remained on the WL, those who subsequently had a DD or LD transplant) for change in satisfaction comparing treatment at 1 year (eg, 1 year post-transplant) with previous renal treatment (eg, dialysis at recruitment) using RTSQc scores.

For the qualitative interview data,[37] thematic analysis based on a pragmatic and critical realist approach was used to explore the experiences of transplant recipients. Field notes were reviewed and transcripts read three times for familiarisation prior to analysis. The coding was completed in Microsoft Word, then entered into NVivo 10 software for analysis. Using established guidelines,[38] independent initial coding by AG established major themes derived from the data which enabled development of a coding framework (AG, CB and MC). There was substantial coder agreement (AG and JB) on five interviews; AG completed the remaining coding on subsequent interviews. Reiteration of responses indicated data saturation had been achieved after 20 interviews.

## RESULTS
### Descriptive findings
Descriptive data for the groups at recruitment (165 patients recruited while on WL for a kidney transplant, 104 DD recipients and 94 LD recipients) are shown in table 1. Almost 13 per cent (12.5%) of DD recipients and 47.5% of LD recipients received their transplants before requiring dialysis. Almost 18 per cent (17.6%) of those on the WL were not yet on dialysis when recruited to the study. At recruitment, 47 (28.5%) patients on the WL had had a previous kidney transplant that failed, while 13 LD (13.8%) and 12 DD (11.5%) recipients had had a previous transplant.

Of all those who received a DD transplant during the study, 80 (55.2%) received a kidney from a donor after circulatory death (DCD), and 65 (44.8%) received a kidney from a donor after brain death (DBD). Sixty-nine LD recipients (57.5%) received a kidney from a relative (35 from parent or adult child, 24 from a sibling and 10 from another relative, eg, aunt, uncle, niece, nephew, cousin), while 51 (42.5%) received a genetically unrelated donor kidney (via the paired/pooled exchange or altruistic donors).

In total, 82 (22.6%) participants did not complete measures at 1 year; 33 (9.1%) were not contactable, 39 (10.7%) did not return questionnaires and 10 were known to have died (2.8%). Responders at 1 year were older ($t_{(120.55)}$=−2.4, p=0.017), and had better EQ-VAS health ratings ($t_{(347)}$= −2.9, p=0.005) at recruitment compared with non-responders. There were no differences in sex, employment status, civil status, education or previous RRT between those who did and did not respond at

1 year. Responders at 1 year were more likely to be white ($\chi^2$=10.3, $df$=4, p=0.016) than non-responders. Missing data were fewer than 5% for all variables and were missing at random, so cases were deleted analysis by analysis.

For those 20 participants who took part in the qualitative interviews: 10 were women; 13 were living with a partner or married, 4 were single and 3 were separated or divorced; 9 were working, 8 were retired and 3 were unemployed/long-term disabled; 6 had had a previous transplant; and 17 were white with the remaining 3 participants being Chinese, Asian or identified as mixed race; 6 were transplanted before requiring dialysis, 12 were previously on haemodialysis and 2 received peritoneal dialysis prior to transplant.

### Quantitative findings
LD recipients were significantly younger compared with the other groups ($F_{(3, 375)}$=9.7, p<0.001; see table 1). The majority of DD recipients and patients on the WL were on haemodialysis, while a larger proportion of LD recipients were transplanted before commencing dialysis ($\chi^2$=23.3, $df$=2, p<0.001). LD recipients were more likely to own a car ($\chi^2$=6.9, $df$=2, p=0.032). Those who remained on the WL differed from those who received a transplant on two marginally related factors ($r$=0.11, p=0.039); patients on the WL were more likely to have experience of a previous transplant that failed ($\chi^2$=16.8, $df$=2; p<0.001) and experience of mental health problems (as recorded by research nurses from medical notes; $\chi^2$=8.9, $df$=2; p=0.011). DD recipients and participants on the WL reported fewer educational qualifications than LD recipients ($\chi^2$=13.7, $df$=6, p=0.033). All subsequent analyses controlled for age, previous RRT, previous experience of a transplant, education, car ownership and history of mental health problems. Participants who were classified as obese at recruitment (BMI >30) or had a diagnosis of diabetes were more likely to experience worse QoL than those without either condition (p<0.05). In separate research conducted as part of the ATTOM programme, obesity was associated with a greater likelihood of delayed graft function in DD transplant recipients, and diabetes was related to a greater risk of transplant failure in LD transplant recipients.[39] Analyses were therefore run controlling for obesity and diabetes. The difference in generic QoL at baseline in those patients who remained on the WL and who did not go on to receive a transplant disappeared, but the sample size was reduced by 16 participants. There were no meaningful differences in the outcomes at 12-month follow-up in the analyses that did and did not control for these factors, so reported findings do not control for obesity and diabetes.

Table 2 shows the correlations between the main outcome variables at 12-month follow-up. Table 3 shows the cross-sectional differences at 1 year post-recruitment/ post-transplant between those who remained on the WL for a transplant (n=98), and those who received a DD (n=145) or LD transplant (n=120). Participants on the WL reported significantly worse scores on all PROMs

**Table 1** Summary of baseline demographic characteristics of groups at recruitment

| Variable | Wait-list group (n=165) M (SE) | DD recipients (n=104) M (SE) | LD recipients (n=94) M (SE) | P value |
|---|---|---|---|---|
| Age* in years | 50.7 (1.6) | 51.1 (1.3) | 43.9 (1.4) | <0.001 |
| Time on waiting list in days | 980 (124.9) | 953 (284.9) | 990 (414.4)† | 0.995 |
| Sociodemographic variables | N (%) | N (%) | N (%) | P value |
| Age* | | | | <0.001 |
| 18–34 | 25 (15.1) | 12 (11.5) | 29 (30.9) | |
| 35–49 | 46 (27.9) | 32 (30.8) | 32 (34.0) | |
| 50–64 | 66 (40.0) | 41 (39.4) | 27 (28.7) | |
| 65–75 | 28 (17.0) | 19 (18.3) | 6 (6.4) | |
| Sex: Female | 59 (35.8) | 40 (38.5) | 34 (36.2) | 0.949 |
| Ethnicity | | | | 0.411 |
| White | 134 (81.2) | 84 (80.8) | 85 (90.4) | |
| Black | 15 (9.1) | 10 (9.6) | 5 (5.3) | |
| Asian | 1 (0.6) | 1 (1.0) | 0 (0.0) | |
| Chinese | 13 (7.9) | 7 (6.7) | 3 (3.2) | |
| Mixed | 2 (1.2) | 2 (1.9) | 1 (1.1) | |
| Marital status | | | | 0.462 |
| Single | 32 (19.4) | 20 (19.2) | 27 (28.7) | |
| Married/living with partner | 98 (59.4) | 63 (60.6) | 60 (63.8) | |
| Divorced/ separated/ widowed | 35 (21.2) | 121 (20.2) | 7 (7.5) | |
| Education | | | | 0.166 |
| No qualifications‡ | 32 (19.4) | 16 (15.4) | 8 (8.5) | 0.033 |
| Basic (GCSE/ A level/NVQ 1–3) | 94 (57.0) | 66 (63.5) | 68 (72.3) | |
| Higher (degree/ higher degree/ NVQ 4–5) | 39 (23.6) | 22 (21.1) | 18 (19.2) | |
| Car ownership: Yes | 134 (81.2) | 89 (85.6) | 90 (95.7) | <0.000 |
| Clinical variables | | | | |
| Primary renal diagnosis | | | | 0.515 |
| Diabetes: Type 1 or 2 | 9 (5.5) | 11 (10.6) | 5 (5.3) | |
| Glomerulonephritis | 50 (30.3) | 32 (30.8) | 42 (44.7) | |
| Interstitial nephritis/ pyelonephritis | 21 (12.7) | 13 (12.5) | 16 (17.0) | |
| Hypertension/ large vessel disease | 10 (6.1) | 8 (7.7) | 0 (0.0) | |
| Cystic/ hereditary/ congenital disease | 47 (28.4) | 19 (18.3) | 14 (14.9) | |
| Other conditions | 28 (17.0) | 21 (20.1) | 17 (18.1) | |
| Comorbid conditions | | | | |
| Diabetes | 20 (12.2) | 16 (15.4) | 8 (8.5) | <0.005 |
| Heart disease | 19 (11.6) | 6 (5.8) | 7 (7.4) | 0.006 |
| Heart failure | 6 (3.7) | 4 (3.8) | 0 (0.0) | 0.305 |
| Liver disease | 4 (2.4) | 3 (2.9) | 1 (1.1) | 0.299 |
| Mental health problems | 14 (8.5) | 2 (1.9) | 5 (5.3) | 0.022 |
| Obesity (BMI >30) | 32 (19.5) | 22 (21.2) | 14 (14.9) | 0.600 |
| Previous renal replacement therapy | | | | <0.001 |
| No dialysis** | 29 (17.6) | 13 (12.5) | 43 (45.7) | |
| Peritoneal dialysis | 27 (16.4) | 23 (22.1) | 14 (14.9) | |
| Haemodialysis (HD)§ | 109 (66.0) | 68 (65.4) | 37 (39.4) | |
| Central venous catheter | 35 (53.1) | 31 (55.6) | 21 (56.8) | |
| Arteriovenous fistula | 22 (33.3) | 28 (41.2) | 15 (40.5) | |
| Unknown | 9 (13.6) | 9 (13.2) | 1 (2.7) | |

Continued

**Table 1** Continued

| Variable | Wait-list group (n=165) M (SE) | DD recipients (n=104) M (SE) | LD recipients (n=94) M (SE) | P value |
|---|---|---|---|---|
| Previous transplant failure¶ | 47 (28.5) | 12 (11.5) | 13 (13.8) | 0.001 |
| Highly sensitised: Yes | 8 (4.9) | 9 (8.7) | 8 (8.5) | 0.842 |
| Donor related to patient: Yes | – | – | 56 (59.6) | |
| Induction suppression | | | | 0.116 |
| None | – | 16 (15.4) | 19 (20.2) | |
| Anti-thymocyte globulin | – | 7 (6.7) | 1 (1.1) | |
| Basiliximab | – | 66 (63.5) | 53 (56.4) | |
| Campath | – | 15 (14.4) | 21 (1.1) | |
| Calcineurin inhibitor maintenance | | | | 0.073 |
| Tacrolimus | – | 98 (94.2) | 93 (98.9) | |
| Ciclosporin | – | 6 (5.8) | 1 (1.1) | |
| Anti-proliferative drugs | | | | 0.341 |
| None | – | 13 (12.5) | 8 (8.5) | |
| Mycophenolate | – | 70 (67.3) | 54 (57.4) | |
| CellCept | – | 8 (7.7) | 16 (17.0) | |
| Myfortic | – | 7 (6.7) | 10 (10.6) | |
| Azathioprine | – | 6 (5.8) | 6 (6.4) | |
| Steroid maintenance plan | | | | 0.193 |
| None | – | 9 (8.7) | 12 (12.8) | |
| Withdraw within 3 months | – | 18 (17.3) | 24 (25.5) | |
| Long-term use | – | 77 (74.0) | 58 (61.7) | |

303132

*LD recipients were younger.
†Figures here reflect only those who were on the WL for a DD transplant before receiving a LD transplant (n=16).
‡Fewer LD recipients reported no educational qualifications.
§More DD recipients and patients on the WL reported being on HD prior to transplantation/listing.
¶WL patients reported more previous transplant failures.
**A larger proportion of LD recipients were transplanted pre-emptively.
BMI, body mass index; DD, deceased-donor transplant; LD, living-donor transplant; M, mean; PROM, patient-reported outcome measure; WL, waiting list group.

at 1 year compared with the two transplant groups (all p<0.001). LD recipients reported better generic QoL, RDQoL, well-being and treatment satisfaction (all p<0.05) than DD recipients. These analyses suggest that LD recipients have better outcomes than DD recipients post-transplant, but do not consider baseline differences in PROMs pre-transplant.

Table 4 and figure 2 depict the prospective analyses comparing over time, those who remained on the WL (n=98) and those who were recruited as patients on the

**Table 2** Correlations between outcome measures at 12-month follow-up for all participants

| | Variable | 1 | 2 | 3 | 4 | 5 | 6 |
|---|---|---|---|---|---|---|---|
| 1 | Generic QoL | – | | | | | |
| 2 | Renal-dependent QoL | 0.499*** | – | | | | |
| 3 | Well-being | 0.740*** | 0.551*** | – | | | |
| 4 | Health status (EQ-VAS) | 0.725*** | 0.415*** | 0.652*** | – | | |
| 5 | Health-utility values | 0.674*** | 0.444*** | 0.678*** | 0.612*** | – | |
| 6 | Renal treatment satisfaction (RTSQs) | 0.545*** | 0.420*** | 0.521*** | 0.454*** | 0.364*** | – |
| 7 | Change in treatment satisfaction (RTSQc) | 0.282*** | 0.134* | 0.292*** | 0.293*** | 0.125* | 0.617*** |

*p<0.05, **p<0.01, ***p<0.001.
EQ-VAS, EuroQoL Visual Analogue Scale; QoL, quality of life; RTSQc, RTSQ change version; RTSQs, Renal Treatment Satisfaction Questionnaire status version.

**Table 3** Summary of adjusted means, standard errors and main effects examining cross-sectional differences in groups in patient-reported outcomes at 1 year, controlling for age, previous renal replacement therapy, previous experience of a transplant, education, car ownership and mental health problems

| | Wait-list group | | DD transplant | | LD transplant | | | | | |
|---|---|---|---|---|---|---|---|---|---|---|
| | Mean (SE) | 95% CI | Mean (SE) | 95% CI | Mean (SE) | 95% CI | F | df | P value | Partial $\eta^2$ |
| Generic QoL*† | 0.4 (0.1) | 0.1 to 0.7 | 1.4 (0.1) | 1.1 to 1.6 | 1.8 (0.1) | 1.5 to 1.9 | 27.6 | 266 | <0.001 | 0.20 |
| Renal-dependent QoL*† | −4.4 (0.3) | −4.9 to −3.9 | −2.7 (0.2) | −3.1 to −2.4 | −1.9 (0.2) | −2.3 to −1.5 | 27.2 | 266 | <0.001 | 0.18 |
| Well-being* | 19.0 (0.9) | 14.3 to 20.8 | 24.9 (0.7) | 23.7 to 26.3 | 27.4 (0.8) | 25.8 to 28.9 | 24.7 | 265 | <0.001 | 0.17 |
| Health status (EQ-VAS)* | 62.4 (2.2) | 58.0 to 66.8 | 77.9 (1.6) | 74.8 to 81.2 | 82.4 (1.9) | 78.7 to 86.1 | 24.1 | 263 | <0.001 | 0.18 |
| Health-utility values* | 0.7 (0.1) | 0.6 to 0.8 | 0.8 (0.1) | 0.8 to 0.9 | 0.9 (0.1) | 0.8 to 0.9 | 13.1 | 259 | <0.001 | 0.11 |
| Renal treatment satisfaction (RTSQs)*† | 58.6 (1.3) | 56.1 to 61.1 | 67.4 (0.9) | 65.5 to 69.2 | 72.6 (1.1) | 70.5 to 74.7 | 34.3 | 265 | <0.001 | 0.21 |
| Change in treatment satisfaction (RTSQc)* | 14.1 (1.7) | 10.8 to 17.4 | 29.2 (1.3) | 26.7 to 31.7 | 30.9 (1.4) | 28.2 to 33.8 | 33.6 | 253 | <0.001 | 0.22 |

Partial $\eta^2$ is a measure of effect size that measures the proportion of the total variance in a dependent variable that is associated with the membership of different groups defined by an independent variable, in which the effects of other independent variables and interactions are partialled out. Partial $\eta^2$ is seen as giving small (0.01), medium (0.09) or large (0.25) effect sizes.
*WL group reported worse outcomes compared with DD or LD recipients (p<0.001).
†LD recipients reported better scores than DD recipients (*p*<0.05).
DD, deceased-donor transplant; EQ-VAS, EuroQoL Visual Analogue Scale; LD, living-donor transplant; M, mean; QoL, quality of life; RTSQc, Renal Treatment Satisfaction Questionnaire change version; RTSQs, RTSQ status version; SE, standard error; WL, waiting list.

WL, but who subsequently received a LD (n=26) or DD transplant (n=41). There were significant interaction effects between the three groups and from recruitment to 1 year post-recruitment/post-transplant for all PROMs. At recruitment, patients on the WL who did not go on to receive a transplant reported worse generic QoL than

**Table 4** Summary of adjusted means, standard errors and interaction effects examining differences in those who remained on the waiting list and those who subsequently had a LD or DD transplant, in patient-reported outcomes from pre-transplant to 12 months post-transplant/recruitment to 12 months, controlling for age, previous renal replacement therapy, previous experience of a transplant, education, car ownership and mental health problems

| | | Generic QoL | Renal-dependent QoL | Well-being | Health status EQ-VAS scores | Health-utility values | Renal treatment satisfaction (RTSQs) |
|---|---|---|---|---|---|---|---|
| Recruitment | | | | | | | |
| WL | M (SE) | 0.39 (0.15) | −4.29 (0.27) | 22.94 (0.78) | 69.32 (2.63) | 0.78 (0.03) | 58.43 (1.76) |
| | CI | 0.10 to 0.69 | −4.82 to −3.75 | 21.40 to 24.48 | 64.10 to 74.53 | 0.73 to 0.83 | 54.94 to 61.93 |
| DD | M (SE) | 1.11 (0.21) | −3.98 (0.39) | 23.18 (1.04) | 67.42 (3.50) | 0.86 (0.03) | 60.10 (2.42) |
| | CI | 0.69 to 1.52 | −4.74 to −3.12 | 21.11 to 25.25 | 60.48 to 74.36 | 0.79 to 0.92 | 55.29 to 64.91 |
| LD | M (SE) | 0.77 (0.27) | −4.08 (0.49) | 22.37 (1.40) | 61.10 (4.27) | 0.86 (0.04) | 54.76 (3.15) |
| | CI | 0.23 to 1.31 | −5.06 to −2.15 | 19.59 to 25.15 | 51.78 to 70.42 | 0.77 to 0.94 | 48.50 to 61.01 |
| 12 months post-recruitment / 12 months post-transplant | | | | | | | |
| WL | M (SE) | 0.39 (0.14) | −4.45 (0.27) | 18.95 (0.99) | 61.89 (2.39) | 0.69 (0.03) | 59.11 (1.45) |
| | CI | 0.11 to 0.68 | −4.99 to −3.91 | 16.99 to 20.92 | 57.16 to 66.63 | 0.63 to 0.75 | 56.24 to 61.99 |
| DD | M (SE) | 1.30 (0.20) | −2.92 (0.39) | 24.45 (1.33) | 77.19 (3.18) | 0.85 (0.04) | 66.07 (1.99) |
| | CI | 0.90 to 1.69 | −3.68 to −2.15 | 21.81 to 27.09 | 70.88 to 83.49 | 0.76 to 0.93 | 62.12 to 70.03 |
| LD | M (SE) | 1.77 (0.26) | −2.41 (0.49) | 27.30 (1.79) | 83.90 (4.27) | 0.93 (0.05) | 74.07 (2.59) |
| | CI | 1.25 to 2.28 | −3.39 to −1.43 | 23.74 to 30.84 | 75.43 to 92.37 | 0.82 to 1.03 | 68.93 to 79.21 |
| | F | 5.91 | 12.08 | 14.43 | 20.63 | 8.04 | 15.25 |
| | df | 103 | 102 | 111 | 110 | 108 | 101 |
| | P value | 0.004 | <0.001 | <0.001 | <0.001 | 0.001 | <0.001 |
| | Partial $\eta^2$ | 0.10 | 0.19 | 0.21 | 0.27 | 0.13 | 0.23 |

DD, those recruited as wait-list patients, but who subsequently had a deceased-donor transplant; EQ-VAS, EuroQoL Visual Analogue Scale; LD, those recruited as wait-list patients, but who subsequently had a living-donor transplant; M, mean; QoL, quality of life; RTSQs, Renal Treatment Satisfaction Questionnaire status version; WL, those who remained on the waiting list from recruitment to 12m post-recruitment.

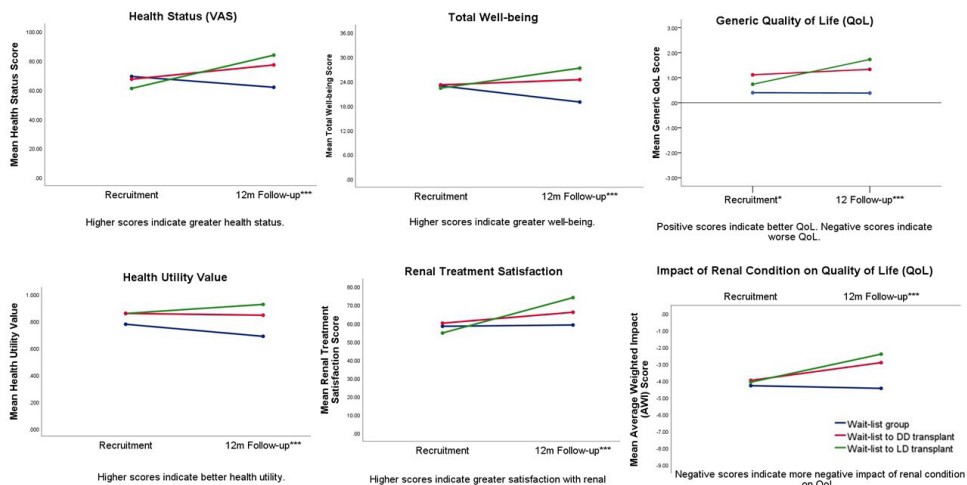

**Figure 2** Interaction graphs showing differences in outcomes at recruitment and at 1 year post-transplant/post-recruitment in those who remained on the WL for a kidney transplant (n=98), or those who were recruited as patients on the WL and subsequently received a DD kidney transplant (n=41), or a LD kidney transplant (n=26). Adjusted scores shown, controlling for age, previous RRT, previous experience of a transplant, education, car ownership and history of mental health problems. *p<0.05 WL vs DD, ***p<0.001 WL vs LD/DD. DD, deceased donor; LD, living donor; m, month; RRT, renal-replacement therapy; VAS, Visual Analogue Scale; WL, wait-list group.

those who went on to receive DD transplants (p=0.028). Patients who remained on the WL reported worsening EQ-VAS health ratings (p=0.004), utility values (p<0.001) and well-being (p<0.001) from recruitment to 1 year, but generic QoL, renal-dependent QoL and renal treatment satisfaction remained stable over time. In contrast, DD recipients reported improvements in EQ-VAS health ratings (p=0.017), renal-dependent QoL (p=0.011) and renal treatment satisfaction (p<0.001), but no change in utility values (p=0.450), well-being (p=0.380) or generic QoL (p=0.200) from pre-transplant to post-transplant. LD recipients reported improvements in all outcomes post-transplant (p<0.01). Though LD recipients reported greater treatment satisfaction than DD recipients (p=0.038), the transplant groups did not differ on any other PROMs post-transplant. Patients still on the WL reported worse scores on all PROMs at 1 year compared with both transplant groups (p<0.001). The effect sizes ranged from small to moderate (table 4; partial $\eta^2$=0.10–0.27). Worse generic QoL post-transplant was reported by 16% of DD recipients and 5% of LD recipients. In contrast, 7% of DD recipients and no LD recipients reported more negatively impacted AWI scores from pre-transplant to post-transplant. Similarly, 8% of DD recipients and no LD recipients reported worse health status as measured by the EQ-VAS from pre-transplant to post-transplant.

Using the treatment satisfaction scores from the RTSQs at recruitment as a further covariate in analyses, both DD (mean=29.6, SE=2.7) and LD recipient groups (mean=31.8, SE=3.4) reported greater satisfaction with renal treatment at 1 year versus previous treatment than those still on the WL (mean=14.4, SE=1.9; $F_{(1, 97)}$=16.1, p<0.001; partial $\eta^2$=0.25; figure 3).

## Qualitative findings

Three themes were identified from the qualitative interviews; participants discussed the positive impact of transplantation, the impact of expectations on the ability to cope post-transplant, and their feelings towards donors. Illustrative examples of quotations can be seen in table 5. Overall, participants viewed their transplant as a way to return to 'normal'. Both transplant groups reported physical improvements, reduced dietary and fluid restrictions, and improved lifestyle post-transplant. Many

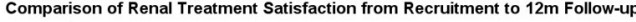
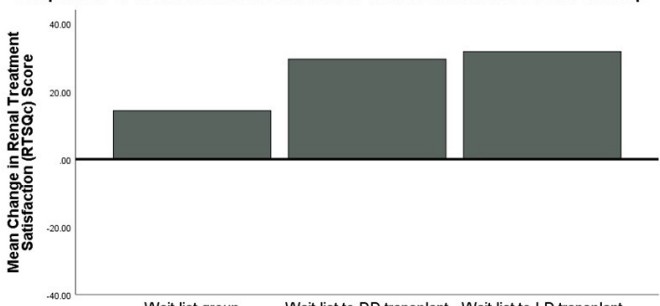

**Figure 3** Bar graph showing differences in satisfaction with current renal treatment compared with previous renal treatment at 1 year post-transplant/post-recruitment in those who remained on the waiting list for a kidney transplant, or those who received a DD or LD kidney transplant after recruitment. Adjusted scores shown, controlling for age, previous RRT, previous experience of a transplant, education, car ownership, history of mental health problems and RTSQs scores at recruitment.
DD, deceased donor; LD, living donor; m, month; RRT, renal-replacement therapy; RTSQs, Renal Treatment Satisfaction Questionnaire status version.

**Table 5** Summary of qualitative themes with illustrative quotations

| Theme | Illustrative quotations |
|---|---|
| Positive impact of transplantation | Physical changes:<br><br>Uh, I feel a hell of a lot better. I was terribly, terribly ill, and I even look like me now. I didn't look like me for quite a long time. I had yellow eyes and grey skin. There's no doubt at all that it's made a massive positive difference. *Woman, non-related-LD transplant following HD*.<br><br>My energy levels are amazing… my mates and that, they just couldn't believe it. They said 'god you look better than I do'. And I was full of life, I'm full of life. That's the sort of euphoria that you need, that gradually, very gradually subsided to normality I suppose. But I was, I was almost hyper. I felt so good. *Man, pre-emptive related-LD transplant*.<br><br>Lifestyle:<br><br>Everything's gone right back to normal now it's fine… You go back to practically normal. *Man, DD transplant following HD*.<br><br>I feel a bit normal again you know yeah *Woman, DD transplant following APD*.<br><br>I'm back to how I was some years back. *Woman, pre-emptive non-related-LD transplant*.<br><br>It gives you more freedom to, to actually live the life that I wanted to live, before dialysis… me and my wife are planning our honeymoon because I missed that because I just hit dialysis after my wedding. So now… after all these years on dialysis, we can plan for the future. *Man, DD transplant following HD*.<br><br>Being able to work is massive for me and I can now work full-time and not only work full-time but own my own home which I wasn't able to do before. *Woman, non-related-LD transplant following HD*.<br><br>I can just get up and do things without worrying… It is so much better than what it was before. Before I thought I was going to die, now ok I know I'm going to die but maybe hopefully in the future. So, I don't worry about the little things anymore. *Woman, DD transplant following HD*.<br><br>I mean I can drink as much as I like now. That before you could only drink like 1 L a day and that. *Woman, DD transplant following HD*.<br><br>I mean I've gone back; I have a normal diet now. *Man, non-related-LD transplant following APD*. |
| Impact of expectations on ability to cope post-transplant | Physical changes:<br><br>I have to drink like a fish. So, I'm all bloated … all I've got to show for (the transplant) is a huge stomach, because obviously they fitted a kidney in and the bag in my bladder so I look like I'm pregnant all the time… I'm a bit of a recluse actually. I just stay in all the time or if I was going out I can't, I can't wear anything fitted anymore because I look pregnant. Just, it's really, rubbish! Woman, *pre-emptive DD transplant*.<br><br>I've lots of marks on my face (from a reaction to the medication) which isn't the greatest thing for confidence in the world. *Man, related-LD transplant following HD*.<br><br>I'm on steroids so I have put on a bit of weight. *Man, pre-emptive DD transplant*.<br><br>There's a thing when you have a transplant that you tend to pile it on, pile the weight on quite badly. *Woman,* non-*related LD transplant following HD*.<br><br>You have to take a lot of water … after your transplant. You're supposed to take 2 L every day…I find I'm running to the toilet a lot. It breaks your sleep at night, you're up maybe three times during the night and you're waking up and you've not really properly slept. *Man, pre-emptive related- LD transplant*.<br><br>I hadn't used my bladder for 7 years, and it was pretty darn painful (to use it again) and took a lot of running to the loo every 5 min. *Woman, DD transplant following HD*.<br><br>Then oh god I was weeing for Britain! You know, it was, that was something I found very hard to get used to because as you know your bladder would have shrunk to nothing … when I had the transplant, I had to drink 6 L. I found that quite hard. *Woman, DD transplant following HD*.<br><br>When you're on dialysis you dream of being able to drink. Once you're told you have to drink its actually really difficult…. It's become normal not to drink anything and then all of a sudden, you're told to drink at least 3 or 4 L of water or fluid a day. *Woman, non-related-LD transplant following HD*.<br><br>I'm still a little bit sort of behind where I would be but I've got like 30, I've got 35% function which is a little bit beyond where I need to be feeling properly sort of strong and active. But I've had a year of it being sort of, it's never been really bad but I've had a year of it being not quite where it needs to be so I'm a lot more tired than I'd like to be. *Man, DD transplant following HD*.<br><br>I've just not got the stamina I had before like you know. *Man, non-related-LD transplant following APD*. |

| Theme | Illustrative quotations |
|---|---|
| | I know it's taken me a whole year to really be as I was before and I still feel in some ways I'm not quite back to what I was before. *Woman, pre-emptive non-related-LD*. |
| | My metabolism seems to have changed a great deal from the moment I had the transplant. *Woman, related-LD transplant following HD*. |
| | I've always wanted to try out cabin crewing before but now I can't even think of doing it because of my health conditions cos I can deteriorate if I start to fly. I have to be very cautious about what I want, you know like what I choose now, just to have to be very careful basically, even holidays. It's just very depressing. *Woman, DD transplant following PD*. |
| Feelings towards donor | Gratitude: |
| | I'm so grateful for having it. *Man, DD transplant following HD*. |
| | I'm very lucky to get the chance, due to my donor like; you know to get a chance of living again, and being without dialysis. *Woman, DD transplant following PD*. |
| | I mean I really, I'm incredibly, incredibly lucky I've got this person who I don't know, somewhere in the world, somewhere in the UK, I don't know where, to thank for this change of my, total change of my life. *Woman, non-related-LD transplant following HD*. |
| | Worry: |
| | Psychologically I would rather have had a dead donor. For the simple reason that you know, it's difficult to deal with taking a part of your daughter's body out which she may need in later years. She was 27 at the time. So… it was very difficult, very difficult. *Man, pre-emptive related-LD transplant*. |
| | She's an only child and she has two children of her own and (it's) such a sacrifice and if anything had gone wrong or even now if anything happens to her and her other kidney, … I worry about her. *Woman, related-LD transplant following HD*. |
| | You worry about the future you know. My (donor) she's very young, she's all of her life and that you know. *Man, pre-emptive related-LD transplant*. |

DD, deceased donor; LD, living donor; HD, haemodialysis; PD, peritoneal dialysis; APD, automated peritoneal dialysis.

DD recipients had not yet returned to work, but some LD recipients reported improvements related to their employment, due to their increased physical ability.

Patient expectations of transplantation had a strong influence on whether or not patients felt their transplant was successful. Many patients believed that once they received a transplant, they would return to the same level of physical health they experienced prior to their kidney problems. Six out of the 10 LD recipients reported ongoing physical problems at 1 year post-transplant, such as reduced stamina, which they did not anticipate. This led to more difficulty adjusting post-transplant, and lower rates of treatment satisfaction. An unexpected side effect of transplantation included changes in physical appearance, such as scarring, bloating from the surgery or weight gain from medication. Patients who experienced these changes found it harder to cope, and their QoL was negatively impacted. Other unexpected experiences, such as difficulty adapting to drinking large quantities of fluids and relearning bladder control post-transplant, were issues that participants found particularly difficult.

Although all participants reported feeling immense gratitude to their donors, LD recipients worried about the health of their donors, leading to feelings of guilt up to 1 year post-transplant. This was particularly evident in three recipients whose donors were their adult children. Continued worry about their donors meant that LD recipients were likely to report difficulties in coping post-transplant.

## DISCUSSION

Previous research has shown consistently that there are differences in the patient characteristics of DD and LD recipients.[14 15 40] Controlling for such factors, our cross-sectional analyses suggest that LD transplantation conveys an advantage over DD transplantation for QoL and treatment satisfaction. However, once baseline differences in factors such as age, education and time on dialysis were controlled for (in the subset of patients with true baseline measures pre-transplant), DD and LD kidney transplant recipients are found to have similar outcomes post-transplant. The only significant difference between transplant groups with true baseline data was in treatment satisfaction 1 year post-transplant with LD recipients having greater treatment satisfaction than DD recipients. Previous research showed that LD recipients reported greater improvements within the first few months post-transplant, but LD and DD recipients did not differ after longer intervals of time.[17] Although the improvements seen in the LD recipients are encouraging, our results suggest that they have worse baseline measures of treatment satisfaction, health status and generic QoL before transplant, so their improvements post-transplant are

more evident. Caution should be exercised when interpreting studies that have not controlled for differences at recruitment and rely on cross-sectional follow-up data alone.

The qualitative interviews show that although participants report improvements to their QoL through a return to 'normal' and improvements in lifestyle, participants still experience ongoing negative impact. Although transplantation is considered a positive and successful treatment, it cannot minimise the negative impact of a renal condition completely. The findings reflect those of previous studies, which report that some transplant recipients find it difficult to adjust and cope post-transplant.[41–43] Differing themes emerged for the LD and DD recipients; Although slightly more LD recipients had returned to work by 1 year (5/10 LD vs 3/10 DD), they were more likely to mention difficulties in adapting post-transplant and feel that their positive expectations for recovery and return to work were not realised. Most LD recipients were not on dialysis before they received their transplant, so the benefits they perceived from transplantation were fewer than those who have experienced dialysis. It is important to manage patient expectations (eg, recovery times) to avoid disappointment and promote effective coping with physical and psychosocial changes following transplant. A small number of LD recipients reported experiencing feelings of guilt about the risks to their donors, as has been found elsewhere.[14 15] Further support may be required for those considering such a transplant, to help anticipate and manage the potential for the feelings of guilt which may include deciding to decline the offer. The impact of transplantation on QoL and other PROMs on both recipients and their donors need to be considered, especially when assessing the comparative benefit of LD versus DD transplantation.

In line with our hypotheses, patients on the WL reported worse health and well-being over time. This may in part be due to the fact that almost 18% of those on the WL were not yet on dialysis when recruited to the study but all required dialysis 1 year later. Some patients may find that their health improves once they commence dialysis, but on average health status declined. However, QoL and treatment satisfaction scores did not change over the course of 12 months in those on the WL despite their deteriorating health status and well-being. This study cannot explain in detail the whole experience of patients on the WL for a transplant, and due to the relatively small sample size and small effect sizes for some of the outcomes, some of the results may not have been adequately powered. The lack of differences between groups over time therefore needs to be considered with caution. Despite this, the results highlight differences between health status and QoL.[20] QoL can be affected by multiple aspects of people's lives with their health being just one aspect, so it is important to distinguish between measures of genuine QoL and health status when selecting and interpreting the results from PROMs. This is particularly the case where the treatment

for long-term conditions such as CKD may benefit health status but may impair QoL.

There are some limitations to this study. The sample of transplant participants with pre-transplant data was relatively small (n=67), as they were part of secondary analyses, some of which may be underpowered, so caution needs to be exercised when drawing conclusions. However, the findings provide insight into how QoL and other patient-reported outcomes change over time in transplant recipients. Only those fluent in English were recruited to ensure that participants could complete the questionnaires: cost of linguistic validation for all the PROMs to be translated into other languages exceeded the ATTOM programme's resources. This meant that, for example, few people of South Asian descent took part in our substudy, despite the fact that approximately 9.4% of transplant recipients recruited to the larger ATTOM research programme were of Asian ethnicity (with the great majority being of South Asian descent). Those interviewed for the qualitative interviews were mainly white. Therefore, the sample is less ethnically diverse than the UK renal population as a whole. Patients who responded at 1 year were more likely to be older, white and have better health ratings at recruitment, so our findings may not be generalisable to younger non-white patients, and may give a more positive view of changes in health over time than if those with worse baseline health status were included. Ten patients were known to have died during the study period and no one reported a failed transplant during data collection, limiting the scope of the findings to those who remained healthy enough to be on the WL, and those with a functioning graft. Despite this, participants to our PROMs substudy were recruited as part of ATTOM from renal units across the UK, reflecting the full ATTOM sample in terms of age, sex, previous transplant and rate of pre-emptive transplantation. Most patients had not returned to work at follow-up; those who had returned to work had varying levels of QoL compared with those who had yet to return to work. The long-term impact of taking immunosuppressants, including the increased risk of cancer and other diseases, may not yet be reflected in participants' QoL and other PROMs within the first year of having a transplant. Longer follow-up, therefore, may be required to show any differences across transplant groups, and the long-term impact of immunosuppressant medication on QoL. The focus of this study was on PROMs, but medical variables such as the number of complications post-transplant which may impact on QoL, were not assessed. Nevertheless, this is one of the few studies that provide any pre-transplant data, allowing us to examine changes over time in transplant recipients and, importantly, to control for baseline differences in PROMs. Additionally, the inclusion of patients on the WL along with recipients of LD and DD in analyses which controlled for group differences, allowed for direct comparison of outcomes for transplant recipients and those still on the WL. The paper highlights the importance of measuring and controlling for pre-existing differences between groups. A strength of this study is that it included both qualitative and quantitative data, which allowed for further understanding of the subjective experience of transplantation. It also shows

that examining a range of PROMs provides a much better understanding of the experience of waiting for or receiving a transplant than is provided by health-status measures alone.

These findings have implications for clinical practice including future monitoring of patient-reported outcomes. Both transplant groups reported better outcomes than those still on the WL for a transplant, but there were few differences in PROMs between groups post-transplant after adjustment for potential confounders. Living donation has certain medical advantages, including that recipients do not have to wait as long for the transplant, but it is also important to acknowledge that living donation does not just affect recipients and their donors physically. Discussion is needed about more than just the medical benefits for the recipient and the medical impact on the donor; PROMs need to be better integrated into the information provided to recipients and donors when transplantation is discussed as a treatment option. Knowing that not all patients report improvements in PROMs and why may help future patients avoid disappointment. Routine collection of PROMs data at time of wait-listing and periodically thereafter is required so that a more accurate picture of changes pre-transplant to post-transplant can be achieved from larger more representative data sets. Careful expectation management, information on recovery time, anticipated physical changes (including scar size) and side-effects may help avoid disappointment and ensure better informed decisions about transplantation options. Nevertheless, both transplant types were, overall, more beneficial than remaining on the WL and may be considered similarly advantageous in improving QoL and other patient-reported outcomes.

**Author affiliations**
[1]Department of Psychology, University of Winchester, Winchester, UK
[2]Health Psychology Research Unit, Royal Holloway University of London, Egham, UK
[3]Health Psychology Research Unit, Health Psychology Research Ltd, Egham, UK
[4]Department of Psychology, Royal Holloway, University of London, Egham, UK
[5]Health Sciences, University of Warwick, Warwick Medical School, Coventry, UK
[6]Statistics and Clinical Studies, NHS Blood and Transplant, Bristol, UK
[7]Edinburgh Transplant Centre, Royal Infirmary of Edinburgh, Edinburgh, UK
[8]Richard Bright Renal Unit, Southmead Hospital, Bristol, UK
[9]Department of Renal Medicine, Newcastle Upon Tyne Hospitals NHS Foundation Trust, Newcastle Upon Tyne, UK
[10]Academic Unit of Primary Care and Population Sciences, Faculty of Medicine, University of Southampton, Southampton, UK
[11]Organ Donation and Transplantation, NHS Blood and Transplant Organ Donation and Transplantation Directorate, Bristol, UK
[12]Department of Surgery, University of Cambridge, Cambridge, UK
[13]NIHR Cambridge Biomedical Research Centre and the NIHR Blood and Transplant Research Unit in Organ Donation and Transplantation, University of Cambridge, Addenbrooke's Hospital, Cambridge, UK

**Contributors** AG contributed to the design of the qualitative study, developed the interview schedule, conducted the interviews, developed the coding framework, coded and analysed the quantitative and qualitative data, drafted the manuscript and edited and approved the final submission. JB managed telephone and postal data collection, coded qualitative data, provided feedback on initial drafts of the manuscript and edited and approved the final submission. MC contributed to the design of the qualitative study, development of the interview schedule and coding framework, provided feedback on initial drafts of the manuscript and edited and approved the final submission. HD contributed to the design of the qualitative study, development of the interview schedule and edited and approved the final submission. RJJ, GCO, RR, CT, PR, WM, JLRF and CD contributed to the design, organisation and conduct of the wider ATTOM programme including data collection for this substudy and provided feedback on the interview schedule, and edited and approved the final submission. JAB, RR, GCO, CJEW, CT, CD and JLRF conceived and designed the ATTOM programme, contributed to the design of the present studies, provided feedback on the interview schedule and edited and approved the final submission. CB contributed to the design of the quantitative and qualitative studies, analysis planning and interpretation of results, development of the interview schedule and coding framework, edited early drafts of the manuscript and edited and approved the final submission. AG and CB act as guarantor, and accept full responsibility for the work and/or the conduct of the study, had access to the data and controlled the decision to publish. The corresponding author attests that all listed authors meet authorship criteria and that no others meeting the criteria have been omitted. We thank the patient advisers for their contributions to the planning of ATTOM and discussions of findings.

**Funding** This article presents independent research funded by the National Institute for Health Research (NIHR) under the Programme Grants for Applied Research scheme (RP-PG-0109-10116). The views expressed are those of the authors and not necessarily those of the NHS, the NIHR or the Department of Health.

**Competing interests** All authors completed the ICMJE uniform disclosure form at www.icmje.org/coi_disclosure.pdf. Professor Watson reports personal fees from GlaxoSmithKline outside the submitted work. Professor Clare Bradley reports grants from NIHR during the conduct of the study, and grants from NIHR and GlaxoSmithKline/ViiV Healthcare, outside the submitted work. CB is the majority shareholder in a company, Health Psychology Research Ltd, which licenses her patient-reported outcome measures, for others to use. These questionnaires include the RDQoL, RTSQ and W-BQ used in the ATTOM programme. CB owns the copyright in all of these instruments and when they are licensed for use by commercial companies in their clinical trials, receives royalties. All other authors declared no competing interests. The results presented in this paper have not been published previously in whole or part, except in abstract format.

**Patient consent for publication** Not required.

**Ethics approval** IRAS project ID: 68259. REC reference 11/EE/0120. AG and CB affirm that the manuscript is an honest, accurate and transparent account of the study being reported; that no important aspects of the study have been omitted; and that any discrepancies from the study as originally planned have been explained. The Corresponding Author has the right to grant on behalf of all authors and does grant on behalf of all authors, a worldwide licence to the Publishers and its licensees in perpetuity, in all forms, formats and media (whether known now or created in the future), to (1) publish, reproduce, distribute, display and store the Contribution; (2) translate the Contribution into other languages, create adaptations, reprints, include within collections and create summaries, extracts and/or, abstracts of the Contribution; (3) create any other derivative work(s) based on the Contribution; (4) to exploit all subsidiary rights in the Contribution; (5) the inclusion of electronic links from the Contribution to third-party material wherever it may be located; and, (6) licence any third party to do any or all of the above.

**Provenance and peer review** Not commissioned; externally peer reviewed.

**Data availability statement** No data are available.

**ORCID iDs**
Andrea Gibbons http://orcid.org/0000-0002-7774-0563
Christopher J E Watson http://orcid.org/0000-0002-0590-4901
Clare Bradley http://orcid.org/0000-0002-4079-0364

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
