## [Reviewer comments · BMJ Open]

ARTICLE DETAILS

TITLE (PROVISIONAL)	Changes in quality of life (QoL) and other patient-reported outcome measures (PROMs) in living-donor and deceased-donor kidney transplant recipients and those awaiting transplantation in the UK ATTOM programme: a longitudinal cohort questionnaire survey with additional qualitative interviews.
AUTHORS	Gibbons, Andrea; Bayfield, Janet; Cinnirella, Marco; Draper, Heather; Johnson, Rachel; Oniscu, Gabriel; Ravanan, Rommel; Tomson, Charles; Roderick, Paul; Metcalfe, Wendy; Forsythe, John L. R.; Dudley, Christopher; Watson, Christopher; Bradley, J Andrew; Bradley, Clare

VERSION 1 – REVIEW

REVIEWER	Dr Rahul M. Jindal, MD, PhD, MBA Professor of Surgery and Global Health Uniformed Services University Bethesda, USA
REVIEW RETURNED	27-Oct-2019

GENERAL COMMENTS	The authors have done a commendable job in carrying out this research and placing it in context. This is an excellent study with clear take home message and limitations are discussed. The references are up to date. Overall, a clear addition to qualitative research in transplantation.
--

REVIEWER	Robert Redfield University of Wisconsin USA
REVIEW RETURNED	31-Oct-2019

GENERAL COMMENTS	I think this is an interesting and important study. I think a very important analysis is missing however. While I agree with the authors that qualitative data like this presented and data on QoL and PROMs is very important to report in kidney transplantation and highlights some important findings, I think it is more powerful if the authors could control in the analysis for some clinical outcomes. I say this because I imagine that the results may be effected if the patients had DGF for example, or complications that required a reoperation or readmission. I would be very interested if the results still hold true if a sub group analysis was performed in patients that did not have any complications verses those that did. Or if patients had DGF verses those that did not, or readmission vs those that did not, or rejection. I am interested in the authors thoughts on this. I think this would strengthen the impact of this study. Regardless I think this is an important and novel study worthy of publication.
--

REVIEWER	istvan mucsi university health network, university of toronto, Toronto, ontario, canada
REVIEW RETURNED	15-Dec-2019

GENERAL COMMENTS	Authors` stated objective is to " examine quality of life (QoL) and other patient-reported outcome measures (PROMs) in kidney transplant recipients and those awaiting transplantation". The topic is very interesting, important and relevant. The manuscript is overall well written and it represents a very valuable and important contributions. Several concerns need to be addressed, however, to further improve the manuscript.  1. Authors will need to adhere to the STROBE guidance more closely when presenting their methods and results. Specifically, patient recruitment needs to be described more clearly and the flow chart will need to be updated a bit to better reflect the actual sampling strategy and process. Sampling frame, sampling strategy and process needs to be clearly described. How were waitlisted patients identified? Were patients with stage 4 and 5 CKD considered? What was the actual sampling frame for DD, LD Tx and WL patients? Specific exclusion and inclusion criteria should be described. Why were WL patients not selected for interviews? 2. Research question(s), hypotheses will need to more specifically formulated. Primary and secondary exposures, outcomes defined, variables defined and described. 3. The analytic plan needs to be carefully described in detail. What was the a priori hypothesis, main analysis? Cross-sectional? longitudinal? What time point? How and why was one year post Tx selected as main analysis. Sample size considerations? MV model building needs to be clearly described and justified. 4. I am not quite sure if I fully understand how transplant recipients are considered for the analysis. How exactly were incident transplant recipients recruited? At what time post transplant? Did patients who completed baseline as WL, also complete BL after they received their transplant? 5. How was "matching" done? How many patients were in the matching frame for consideration? What parameters were used for matching? 6. Transplant and dialysis specific information about participants should be provided. A conventional Table 1. with BL characteristics should be provided. In current Table 1 WL time seems to be similar between LD and DD recipients. How is this possible? How did time on dialysis compare? What information does time on WL carry as opposed to time on dialysis? 7. MV model building needs to be better described and justified. How and why did authors select the co-variables for their models? How was comorbidity assessed? Why was it not adjusted for? Clinical variables (kidney/graft function, immunosuppressive treatments (induction and maintenance), history of rejection, hemoglobin level, etc. should be considered, at least presented.
--

	8. The main analysis needs to be justified. If I understand correctly one year post transplant was selected and the groups are formed from participants recruited as incident transplant and participants received a transplant during the follow-up. Therefore the WL group is somewhat unusual, since it represents patients who are left out, at least during the follow up period. This approach has to be strongly justified as it seems a bit unusual. Cross sectional analysis could have considered the baseline group data and longitudinal analysis could have looked at subsequent events. 9. How long after transplant surgery was the baseline data collection done? What was the length of stay, rate and type of complications in the "incident" transplant group? 10. No change in utility, well-being or generic QOL is noted for DD KT. What is the observed power? Is this a measurement issue? This is not even mentioned in the discussion. 11. Authors discuss changes in QOL and PROMs in the WL group. What is the power of their analysis? Is this a question this study can answer? Having advanced CKD and transitioning to RRT is a very complex and difficult process, and interpretation of the quantitative data presented needs to be more cautious. Also, how this demonstrates differences between health status and QOL and how those differences are reflected in the actual measures used, is not clear from the discussion. 12. Authors mention in discussion " One-year follow-up may be too soon to show differences across transplant groups. " In what way do you suggest outcomes may change thereafter? Do you suggest that pre-transplant and transplant related factors will remain the main factors contributing to PROMs post transplant, as opposed to medical and ongoing psycho-social factors? Minor: 1. I suggest to defined the terminology used: QoL, health status and PROM. These are reasonably complex terms and used in various meanings across the literature. This could be useful even for the readership of a general medical journal. 2. Also suggest to use less technical terms in describing results. Seeing proportions, averages and distribution is probably more informative for the general readership than seeing technical terms like F values or Chi2.
--	--

VERSION 1 – AUTHOR RESPONSE

Reviewer(s)' Comments to Author:

Reviewer: 1

The authors have done a commendable job in carrying out this research and placing it in context. This is an excellent study with clear take home message and limitations are discussed. The references are

up to date. Overall, a clear addition to qualitative research in transplantation.

Response: Thank you for your comments

Reviewer: 2

I think this is an interesting and important study. I think a very important analysis is missing however. While I agree with the authors that qualitative data like this presented and data on QoL and PROMs is very important to report in kidney transplantation and highlights some important findings, I think it is more powerful if the authors could control in the analysis for some clinical outcomes. I say this because I imagine that the results may be affected if the patients had DGF for example, or complications that required a reoperation or readmission. I would be very interested if the results still hold true if a sub group analysis was performed in patients that did not have any complications verses those that did. Or if patients had DGF verses those that did not, or readmission vs those that did not, or rejection. I am interested in the authors thoughts on this. I think this would strengthen the impact of this study. Regardless I think this is an important and novel study worthy of publication.

Response:

We agree that including analyses using clinical outcomes would be useful, but the focus of this study was on PROMs and we do not have access to data regarding delayed graft function. We have included the lack of clinical outcomes such as these as a limitation in the discussion.

We did, however, re-run our analyses to see whether any factors that we do have data for may make delayed graft function (DGF) more likely. For example, Wu et al (Transplantation 2019) found that obesity was related to a higher rate of DGF in DD transplant recipients within a separate work-stream of the ATTOM programme. The same study also found that having a diagnosis of diabetes was also related to a greater risk of transplant failure in LD recipients. We re-ran our statistics controlling for obesity (classified as a BMI of >30 at recruitment) and diabetes. We reported in the original manuscript you reviewed that, at recruitment, those patients who remained wait-listed and who did not go on to receive a transplant reported worse generic QoL than those who went on to receive DD transplants ($p=0.028$). When we controlled for obesity and diabetes, this difference disappears, though it does not lead to any meaningful change in any of the results post-transplant, but it does reduce our sample size by 16 participants because of missing data. We have decided to keep the original reported findings, but we have added into the manuscript that we also ran analyses controlling for obesity and diabetes, which did not change the outcomes.

Reviewer: 3

Authors` stated objective is to " examine quality of life (QoL) and other patient-reported outcome measures (PROMs) in kidney transplant recipients and those awaiting transplantation". The topic is very interesting, important and relevant. The manuscript is overall well written and it represents a very valuable and important contributions. Response: Thank you for your comments

Several concerns need to be addressed, however, to further improve the manuscript.

1. Authors will need to adhere to the STROBE guidance more closely when presenting their methods and results.

Specifically, patient recruitment needs to be described more clearly and the flow chart will need to be updated a bit to better reflect the actual sampling strategy and process.

Sampling frame, sampling strategy and process needs to be clearly described. How were waitlisted patients identified? Were patients with stage 4 and 5 CKD considered? What was the actual sampling frame for DD, LD Tx and WL patients? Specific exclusion and inclusion criteria should be described. Why were WL patients not selected for interviews?

Response:

We have now used both the STROBE guidelines and SRQR checklist to describe our study better.

We have also reformatted Figure 1 into a CONSORT flow diagram, so that it is easier to understand and includes the sampling process used.

Some of the inclusion and exclusion criteria were included in the paper, but we have ensured that we have now included all of the inclusion/exclusion criteria in the text. Patients with stage 5 CKD only were recruited to the ATTOM programme; this has now been added in to the text for clarity.

WL patients were identified from the UK Transplant Registry database. Although we did interview people from the waiting list, we did not include these data in this paper; as the experiences of patients on the WL did not illuminate the experiences of those who had transplants and we felt it important to keep the paper focused on the issues in hand. This has been clarified in the text.

2. Research question(s), hypotheses will need to more specifically formulated. Primary and secondary exposures, outcomes defined, variables defined and described.

Response:

The objectives, hypotheses, and definition of outcomes and variables have been further clarified and elaborated in the text.

3. The analytic plan needs to be carefully described in detail. What was the a priori hypothesis, main analysis? Cross-sectional? longitudinal? What time point? How and why was one-year post Tx selected as main analysis. Sample size considerations? MV model building needs to be clearly described and justified.

Response:

We have provided more detail of the analysis plan and have completed the STROBE checklist, which covers this information. For example, in the Methods section, we have identified our analyses (primary and secondary analyses). Further information has been provided regarding hypotheses, the sample size, as well as the timing of the analyses.

4. I am not quite sure if I fully understand how transplant recipients are considered for the analysis. How exactly were incident transplant recipients recruited? At what time post-transplant? Did patients who completed baseline as WL, also complete BL after they received their transplant?

Response:

Incident transplant recipients were recruited by research nurses within one month of receiving a transplant. They were recruited in a quasi-random manner, with the first eligible patient for each group seen each month (November 2011 to March 2013) by each nurse invited to take part.

If patients received a transplant during the follow-up period, they were asked to complete the measures 12 months post-transplant; they were not asked to complete the measures shortly after transplant, as those recruited following transplant had been, as this measurement point was not useful as a 'baseline' as health status and quality of life was impaired by the recent transplant.

Measures completed pre-transplant while wait-listed provided true baseline measures of PROMs before transplant for comparison with PROMs completed 12 months after transplant. These participants were compared with those still wait-listed for a transplant in analyses that allowed for pre- and post-transplant scores to be considered.

All of this information has been added and/or clarified in the Methods section; Study design, participants and procedure. The inclusion of the CONSORT diagram helps to clarify who and when participants were included in various analyses.

5. How was "matching" done? How many patients were in the matching frame for consideration? What parameters were used for matching?

Response:

Prevalent listed patients were selected as matched controls for recruited transplant recipients automatically from the UK Transplant Registry database on a fortnightly basis and were matched for: age (within 5 years), time on the list, and whether they were pre-emptively listed/transplanted or on dialysis. This information has been clarified in the methods section. The CONSORT diagram provides further information as to how many participants were eligible at each stage.

6. Transplant and dialysis specific information about participants should be provided. A conventional Table 1. with BL characteristics should be provided. In current Table 1 WL time seems to be similar between LD and DD recipients. How is this possible? How did time on dialysis compare? What information does time on WL carry as opposed to time on dialysis?

Response:

We have edited Table 1 so that it is more in line with a BL characteristics table. Specifically, we have edited it so that it reflects the sample as a whole at recruitment, and no longer focuses on the sample characteristics at follow-up, as it did previously.

We have added all the information we have about transplant and dialysis of our participants into Table 1 and the main text. WL time is similar across the groups because it was one of the parameters for matching. WL time provides information about how long people have been waiting with an expectation for a kidney transplant. Unfortunately, we do not have information about how long participants were on dialysis for. However, we do have information about what type of RRT participants were receiving, and the LD recipients were more likely to be transplanted pre-emptively, so fewer had any experience of dialysis.

7. MV model building needs to be better described and justified. How and why did authors select the co-variables for their models? How was comorbidity assessed? Why was it not adjusted for? Clinical variables (kidney/graft function, immunosuppressive treatments (induction and maintenance), history of rejection, hemoglobin level, etc. should be considered, at least presented.

Response: We have added in further rationale and explanation of our analyses.

Variables were selected for inclusion as covariates in the analyses if they demonstrated differences between the groups (e.g. age, type of RRT), and/or demonstrated that they had an effect on the outcomes. This controlled for any potential confounding effect they may have had on our findings. The clinical variables described (kidney graft function, immunosuppressive treatments (induction and maintenance) were not all collected at follow-up in our research. We have included information about immunosuppressive treatments recorded at recruitment but we have no data as to whether these treatments changed over time. However, we acknowledge that comorbidity may need to be controlled in analyses, considering variables such as obesity are related to higher rates of delayed graft function. Comorbidity was assessed by the recruiting nurses; any conditions from a pre-specified list mentioned in patients' records when recruited were identified and classified into groups (heart disease, heart failure, liver disease, diabetes, mental health problems, PVD). Height and weight were recorded so that BMI could be calculated.

We investigated our data further in response to your suggestions and have identified that obesity and diabetes are associated with generic QoL. We re-ran our analyses, controlling for diabetes and obesity, but found that they did not meaningfully change the findings, except to reduce our sample size. We have kept the original reported findings in the paper, but we do report that we ran the analyses controlling for diabetes and obesity, and that this did not change the results.

8. The main analysis needs to be justified. If I understand correctly one-year post transplant was selected and the groups are formed from participants recruited as incident transplant and participants received a transplant during the follow-up. Therefore, the WL group is somewhat unusual, since it represents patients who are left out, at least during the follow up period. This approach has to be strongly justified as it seems a bit unusual. Cross sectional analysis could have considered the baseline group data and longitudinal analysis could have looked at subsequent events.

Response:

One-year follow-up was selected as part of the ATTOM programme design; and reflects the fact that in this research, follow-up from transplant is commonly measured at one-year post-transplant.

The main analyses focused on the data from all participants at one-year follow-up; participants were divided based on whether they were still wait-listed (n=98), or had had a DD (n=145) or LD transplant (n=120). Previous research assessing differences in these groups is cross-sectional and this

approach was taken, so that a comparison to previous research could be made.

The secondary analyses focused solely on those originally recruited as WL patients, and were divided into three groups – those who remained wait-listed (n=98), those who received a DD (n=41), and those who received a LD (n=26) transplant. Therefore, all WL patients were included in the analyses. The period of time that they had been wait-listed varied from 15 days to 23 years at baseline. It is true that for those who were still wait-listed at 12 months the minimum time they had been wait-listed was 12 months and some had been wait-listed for more than 2 years (range = 1-21 years; mean = 2 years). Although there is overlap in the range of WL times for transplant recipients who were recruited as WL patients and those still wait-listed they are no longer matched in this respect. However; the findings do not change when we include wait listing time as a covariate, so we do not feel that it is necessary for it to be included in these analyses. We have clarified the main analyses so that it is clearer in the text.

9. How long after transplant surgery was the baseline data collection done? What was the length of stay, rate and type of complications in the "incident" transplant group?

Response: Data collection began at recruitment, which for those who were recruited as transplant recipients, was around the time of the transplant with most being recruited while still in hospital, although the average time was one-month post-transplant. Of the PROMs, only the EQ-5D and Well-being Questionnaire were completed at recruitment. The quality of life and treatment satisfaction measures were completed three months post-transplant once patients had had an opportunity to experience their transplant and associated medication and something of the impact on their lives. The PROMs were all completed again 12m post-transplant.

Unfortunately, we do not have any data on the length of stay, rate, and type of complications in either of the incident transplant groups. However, we re-ran the analysis and controlled for comorbid conditions of obesity and diabetes, but found their inclusion had no meaningful effect on the outcomes. We should note that only those with functioning grafts completed measures at follow-up, so the findings relate only to those with functioning grafts. This is noted as a limitation in the Discussion.

10. No change in utility, well-being or generic QOL is noted for DD KT. What is the observed power? Is this a measurement issue? This is not even mentioned in the discussion.

Response:

We acknowledge that the observed power was not included in the original paper; we felt it more important to report the effect size (which was included), as it identifies the magnitude of the difference between groups, and observed power is calculated in part from the effect size. The achieved power varied across outcomes for the DD KT recipients. Those outcomes that reported a larger effect size also reported greater observed power. However, we did not achieve 80% power in any of the statistics when looking at the DD KT over time, so we do acknowledge that our findings may not be adequately powered. We have made it more explicit in our discussion that we need to exercise caution when interpreting the results.

11. Authors discuss changes in QOL and PROMs in the WL group. What is the power of their analysis? Is this a question this study can answer? Having advanced CKD and transitioning to RRT is a very complex and difficult process, and interpretation of the quantitative data presented needs to be more cautious. Also, how this demonstrates differences between health status and QOL and how those differences are reflected in the actual measures used, is not clear from the discussion.

Response:

We acknowledge that this study alone cannot answer the question of how patients transition from advanced CKD to RRT. Instead, we can report that of our sample of 98 wait-listed patients, over the course of the first year on the waiting list for a transplant, QoL and treatment satisfaction (as measured by the questionnaires we employed) did not deteriorate. However, well-being and health

status did worsen over the same one-year period. We have added these points into the discussion. For the analyses examining those who remain on the waiting list over time, we have adequate power for well-being and health status but not for generic and renal-specific QoL and treatment satisfaction. We acknowledge that some of the findings may be underpowered. As stated in point 10, we have added into the discussion an acknowledgement that some of the analyses are underpowered, so findings should be interpreted with caution.

If health and QoL were similar, then the results might be expected to show the same pattern. Quality of life is a concept that includes multiple aspects of people's lives; health is just one of them. This difference between health status and QoL has been made more explicit in the text, and we have included definitions of both in the Introduction to make this clearer.

12. Authors mention in discussion " One-year follow-up may be too soon to show differences across transplant groups. " In what way do you suggest outcomes may change thereafter? Do you suggest that pre-transplant and transplant related factors will remain the main factors contributing to PROMs post-transplant, as opposed to medical and ongoing psycho-social factors?

Response:

Most patients had not returned to work at follow-up, and the effect of immunosuppressant medication on QoL and other PROMs may not be apparent until after one-year post-transplant. Therefore, we feel that longer follow-up may be required to show subsequent differences across transplant groups, as well as the long-term impact of immunosuppressant medication on QoL. This point has been added and clarified in the discussion.

Minor:

1. I suggest to defined the terminology used: QoL, health status and PROM. These are reasonably complex terms and used in various meanings across the literature. This could be useful even for the readership of a general medical journal.

Response: We have now included clear definitions of PROMs, health status, and QoL into the Introduction.

2. Also suggest to use less technical terms in describing results. Seeing proportions, averages and distribution is probably more informative for the general readership than seeing technical terms like F values or Chi2.

Response:

Although we acknowledge that less technical terms will make it more readable to a wider audience, we feel that it is important to keep reference to the specific statistical tests we used. The description of all statistical tests used is one of the recommendations in the STROBE checklist, so we have kept these terms in the paper. However, Tables 2 and 3 do include information such as confidence intervals of the findings. We hope this is acceptable.

VERSION 2 – REVIEW

REVIEWER	Istvan Mucsi University Health Network, Toronto, Ontario, Canada
REVIEW RETURNED	07-Jul-2020
GENERAL COMMENTS	Authors attempted to answer comments and address concerns. However, I am not at all more clear about research objectives, research questions, design, methods and some of terminology used, than before.

	1. Main objective: "to examine QoL and other PROMs in recipients of DD or LD transplants," Authors hypothesize that "there would be very few differences in outcomes between DD and LD recipients" This hypothesis mentioned 2nd, although it is related to main objective. Rationale and justification for the question and hypothesis is not provided convincingly. The 2nd para on p6 is quite confusing, not clear what the bottom line could be and how that relates to the main question/objective. 2. I still have difficulties understanding the main concepts: PROM, well-being and QOL. Do authors assess general QoL or health related QoL? Why would it be necessary to assess QoL and various PROMs? What is the added information? What are the specific hypotheses associated with each outcomes? 3. Authors claim only moderate correlation between the various constructs mentioned. Would be useful to show those. 4. At the end, it appears that the main objective a xsectional comparison between DD and LD recipients - within a small sample, relatively poor set of covariables to adjust for. Is this worthwhile? Can this question be answered accurately within the limitations of the dataset? 5. a 2nd objective is to compare them (DD and LD) over time with those on the waiting list (WL) for a DD transplant in a matched cross-sectional cohort and in a subsample longitudinally from pre- to post-transplant. This is still very confusing. Cross sectional comparison between WL and KT recipients - is that meaningful? What is the novelty. Is this appropriate comparison? With insufficient clinical documentation? 6. The study design is very difficult to understand still. How is this quasi randomized? What do you mean automatic matching? How is 165 WL matched to 104 DD and 94 LD? How can age be so different if that is one of the matching variable? What was the matching frame? 7. The qualitative and quantitative findings don` t seem to complement each other organically. 8. Apparently participants completed treatment QoL and satisfaction 3 mo post tx. The quality of life and treatment satisfaction measures were completed three months post-transplant once patients had had an opportunity to experience their transplant and associated medication and something of the impact on their lives Authors are concerned that recipients are not stable enough at 1 year post transplant. How was this 3 month timeframe selected then? Also, in a response to previous questions author state: If patients received a transplant during the follow-up period, they were asked to complete the measures 12 months post-transplant; they were not asked to complete the measures shortly after transplant, as those recruited. I see a bit of contradiction here. 9. Arguments about why outcomes assessed at 1 yr post - transplant is not convincing.
--	--

	Authors state: One-year follow-up was selected as part of the ATTOM programme design; and reflects the fact that in this research, follow-up from transplant is commonly measured at one-year post-transplant. This does not sound like a strong rationale. At the same time: Authors mention in discussion " One-year follow-up may be too soon to show differences across transplant groups. ". Why was 1 yr selected then? Further, in their response authors state: Most patients had not returned to work at follow-up, and the effect of immunosuppressant medication on QoL and other PROMs may not be apparent until after one-year post-transplant. KT recipients stabilize clinically at about 3-6 month post transplant. At 1 yr, several of the measured characteristics are stable. 10. Co-variable selection should be done on theoretical basis not based on association in the given sample. 11. one of the hypotheses mentioned states : It was hypothesized that patients still on the WL would have more negative scores on PROMs than those who received a transplant Does this hypothesis still need an analysis? Is this a question? 12. It appears that the sample practically did not include South Asian participants. How did this happen? 13. Authors state that qualitative study participants were representative of the larger samples - how was that assessed and achieved? Is that necessary for qualitative research? 14. In discussion authors state: However, once baseline differences in factors such as age, education, and time on dialysis are controlled for in the subset of patients with true baseline measures pretransplant, DD and LD kidney transplant recipients are found to have similar outcomes post- transplant. in a response to previous question they stated: Unfortunately, we do not have information about how long participants were on dialysis for. 15. To a question about adjusting for additional variables authors state: "We agree that including analyses using clinical outcomes would be useful, but the focus of this study was on PROMs and we do not have access to data regarding delayed graft function." If the research question is PROM/QoL 1 year post-transplant, when designing data collection one would likely want to collect data likely associated with those outcomes. Post-transplant events (length of hospital stay, complications, rejection, DGF are such variables) since they have been repeatedly shown to be associated with additional post-transplant outcomes.
--	---

VERSION 2 – AUTHOR RESPONSE

Reviewer 3 comments

Authors attempted to answer comments and address concerns. However, I am not at all more clear about research objectives, research questions, design, methods and some of terminology used, than before.

We are sorry to hear that the objectives, research questions, design, methods and some of terminology used were not clear. We attempted to clarify these in the previous revisions, for example adding in definitions of the concepts (QoL, PROMs), as well as much more detail of the methodology, but have clarified these further now we have fewer word count constraints; please see below.

1. Main objective: "to examine QoL and other PROMs in recipients of DD or LD transplants,"

Authors hypothesize that "there would be very few differences in outcomes between DD and LD recipients". This hypothesis mentioned 2nd, although it is related to main objective. Rationale and justification for the question and hypothesis is not provided convincingly. The 2nd para on p6 is quite confusing, not clear what the bottom line could be and how that relates to the main question/objective.

The first objective does relate to our second hypothesis, which states that controlling for potential confounding factors, there would be very few differences in outcomes between DD and LD recipients. This is the main point, as not all studies have controlled for such group differences. We have switched the order of the hypotheses so that they align with the order of the listed objectives to make it clearer (see last pph of the Introduction in the manuscript).

In terms of the rationale, we have made this clearer (Introduction, pph 3 onwards). To summarise, of the few studies that have compared LD and DD recipients, they suggest that LD transplantation has an advantage. However, this is not consistent when looking at the various outcomes measured including level of guilt towards their donor or the health and QoL costs to the donor. Importantly, the studies have focused on differences in health status in recipients of living-donor kidneys versus deceased-donor kidneys, which can be confounded by differences in such factors as age in the groups prior to transplant. Living-donor recipients are younger and more likely to receive a pre-emptive transplant without spending time on dialysis: differences which in themselves are likely to result in better health status. Such differences need to be controlled for adequately to assess true differences between the groups based on transplant type. From this, we hypothesised that once variables that may be confounders are controlled for, there will be few differences between the LD and DD recipients.

We have edited pph 3 of the Introduction so that the rationale is clearer for why we included wait-listed patients and why it is important to control for pre-transplant data 'When cross-sectional data are analysed, it is important to have well-matched controls but in practice that is difficult to achieve because those who are left on the waiting list for deceased-donor kidneys tend to have more health problems than those who receive a transplant. Studies that have controlled for underlying differences across the groups, such as differences in age or comorbid disease, report similar outcomes for both transplant groups on health-status measures such as the SF-36,³² although many LD recipients report experiencing feelings of guilt about the risks to their donor,^{14,15} Longitudinal studies, which can include and control for baseline measures of outcomes before participants receive a transplant, suggest that improvements in health outcomes such as the SF-36 can be seen in the first few months post-transplant for both DD and LD, but that these improvements remain stable after this time.¹⁶⁻¹⁹ However, these studies lack any group with which to compare transplant recipients, such as those still awaiting transplantation, and not all include data pre-transplant. Those who are transplanted are more likely to be in better health than those patients still wait-listed for a transplant, so it is important to control for baseline differences between participants who receive a transplant and those who do not.'

2. I still have difficulties understanding the main concepts: PROM, well-being and QoL. Do authors assess general QoL or health related QoL? Why would it be necessary to assess QoL and various PROMs? What is the added information? What are the specific hypotheses associated with each outcomes?

We acknowledge that many find it difficult to grasp the differences between PROMs and the literature frequently confuses the measures, wrongly labelling as 'QoL' or 'health-related QoL' measures that are actually health-status tools. We attempted to clarify these concepts in pph 4, of the Introduction, by providing definitions of the concepts but conscious of the word limitations we were being concise. It seems we were too concise. We elaborate here and in the manuscript Introduction: PROMs: measures of outcomes that are directly reported by the patient, and are usually questionnaires that can be generic or condition-specific. PROMs may measure a wide range of outcomes including health status, QoL, treatment satisfaction, well-being, and other outcomes such as symptoms (Introduction, pph 1).

Health status: aspects of a person's life such as their physical ability, daily functioning, and experience of symptoms. Health-related QoL is a term that is commonly and misleadingly used to refer to health-status tools such as the SF-36 and EQ-5D. If a tool was to measure health-related QoL then the respondent would need to be asked about the impact of their health problems on their QoL. The SF-36 and EQ-5D make no mention of QoL, health-related or otherwise, and are eliciting reports of health status and function without questioning the consequences for QoL (Introduction, pph 4). We do not use of the term health-related QoL except when quoting the words of other researchers and explaining the problems with the way this term has been used.

Quality of life (QoL): QoL is defined by the WHO as an individual's perception of their position in life in the context of the culture and value systems in which they live and in relation to their goals, expectations, standards and concerns and they attempt to measure this with the WHOQOL generic measure of QoL which has various condition-specific additions. However, that measure does not allow for individual differences in the importance of the different components measured or indeed for the possibility that some aspects of life measured in the questionnaire are irrelevant for some respondents. In Bradley's RDQoL used in ATTOM the definition of QoL included in the instructions of the questionnaire is simply 'how good or bad you feel your life to be' and the questionnaire and its scoring allows for individual differences in the relevance of different aspects of life. It also allows for individual differences in the importance of relevant aspects of life for QoL as well as measuring the impact of the renal condition on each relevant aspect of life. On reflection we feel it would be clearer to specify only the definition of QoL included in the RDQoL individualised measure used and we have removed mention of the WHO definition which has not been helpful here (see Introduction, pph 4).

The ADDQoL, RDQoL and other -DQoL measures could reasonably be called 'condition-specific health-related QoL measures' but we don't because the term 'health-related QoL' has been so abused in the literature that the unique qualities of the -DQoL measures would be overlooked if we were to call them 'health-related QoL measures'. These points are made in Bradley (2001) that was referenced in our manuscript.

We measured health status (using the EQ-5D), as well as genuine measures of QoL (generic quality of life (RDQoL overview item)) and condition-specific quality of life (RDQoL Average Weighted Impact (AWI) score indicating the impact of the renal condition on QoL)). The EQ-5D includes a single-item measure of health status in the form of the visual analogue scale that resembles a thermometer (the

EQ-VAS) as well as questions about five more specific aspects of health which are used in calculating utilities used by NICE to determine whether treatments are sufficiently cost-effective to be offered on the NHS. Unfortunately, in relying on a health-status tool, which they frequently mislabel as QoL, NICE risk overemphasising health status in judging cost-effectiveness while failing to recognise the importance of protecting QoL if a treatment is to be viewed as a success.

Understanding the differences between health status, QoL and the impact of the renal condition on QoL and the PROMs used to measure them allows the reader to appreciate the importance and novelty of the paper, as well as to distinguish the very real differences between the variables, so that we identify the differences or similarities across transplant groups, without using inappropriate measures. For example, previous research that claimed to measure quality of life only measured health status and often inappropriately drew conclusions about QoL. In ATTOM we measured both health status with the EQ-5D and QoL (both generic and condition-specific QoL) with the RDQoL. The W-BQ12 was added to provide further detail on the mood states that may well be impaired while on dialysis and the RTSQ measure of treatment satisfaction was included because sister measures of treatment satisfaction for other conditions are known to be particularly sensitive to changes of treatment. It is of particular note that the measure of utilities obtained from EQ-5D scores and relied upon by NICE to determine which treatments to offer on the NHS, did not improve from baseline to 1-year follow up in recipients of DD transplants who had been recruited as wait-listed patients and nor did their generic QoL or well-being improve over that time while their self-reported health (EQ-VAS), renal-dependent QoL and treatment satisfaction did improve significantly in the course of that first year. These findings alone illustrate the importance of using condition-specific measures as well as the need to distinguish between the various health-status measures and genuine quality-of-life measures. We hope to raise awareness of the importance of differentiating between different PROMs and to ensure that any and all benefits and disadvantages of transplantation are measured.

The specific hypotheses related to each outcome are all similar, namely that treatment satisfaction, QoL (generic and condition-specific), health status, and well-being would be better in transplant recipients than in those remaining on the waiting list. Controlling for confounding variables, we hypothesised that there would be no differences between LD and DD recipients.

3. Authors claim only moderate correlation between the various constructs mentioned. Would be useful to show those.

We stated in the introduction that 'Quality of life and health status are only moderately correlated with one another,²⁰ though have removed this generalisation in the latest edits as it depends on the measure of quality of life used. Instead in pph 4 of the Introduction referring to the RDQoL measure we state that 'A single item to measure generic QoL is also included in the measure and this item can be expected to be more strongly related to health status generally than the condition-dependent AWI score.²⁴ We did not refer to correlations between our variables at all in the manuscript. We have, however, included the correlations between our outcomes in Table 2. As you can see in Table 2, although generic QoL is correlated highly with health status ($r=0.725$ with VAS and $r=0.674$ with utility values), renal-dependent QoL (as measured by the RDQoL AWI scores), is only moderately correlated with health status ($r=0.415$ with VAS and $r=0.444$ with utility values).

4. At the end, it appears that the main objective a xsectional comparison between DD and LD recipients - within a small sample, relatively poor set of covariables to adjust for. Is this worthwhile? Can this question be answered accurately within the limitations of the dataset?

This is certainly worthwhile. The sample is 265 transplant recipients including 145 DD and 120 LD recipients: these are not small samples and it is not clear why the Reviewer says this or described the covariates as 'a relatively poor set of covariables to adjust for'. As stated in the 'Analysis' subsection of the Methods section, 'Clinical factors such as previous renal replacement therapy, age, and indicators of SES (such as car ownership and qualifications), can colour a person's perspective on their QoL and treatment satisfaction. We therefore assessed these variables and controlled for variables in the analyses if differences between groups were identified on those variables (using Chi Squared tests or ANOVAs).' Many other variables were considered as potential covariates but did not meet these criteria.

No other studies have included genuine measures of QoL and few have controlled for confounding variables. It is often assumed that we know that there is quality of life benefits of having a LDKT rather than a DDKT, but the measurements used, up to this point, have been flawed, and this has led to a general misunderstanding about what QoL is and resulted in unjustified pressure on patients' families to donate their kidneys.

We acknowledge that the study has limitations; notably the cross-sectional design involved in the analyses of the full sample of 265 transplant recipients and the relatively small size of the subsample (165 wait-listed recruits, 41 of whom went on to have a DD transplant and 26 to have a LD transplant) for whom we have PROMs data pre- and post-transplant. We agree with Reviewers 1 and 2 that the study is important and well worth publishing, as no other study has used a genuine measure of QoL and the impact of the condition on QoL together with health status and other PROMs. When presenting the findings to the British Transplant Society, the results provoked much debate and a wish by transplant surgeons and nephrologists to use PROMs clinically to see if our findings can be replicated with other samples including their own patients. They were keen to improve their understanding of the effects of their interventions on QoL.

5. a 2nd objective is to compare them (DD and LD) over time with those on the waiting list (WL) for a DD transplant in a matched cross-sectional cohort and in a subsample longitudinally from pre- to post-transplant. This is still very confusing. Cross sectional comparison between WL and KT recipients - is that meaningful? What is the novelty. Is this appropriate comparison? With insufficient clinical documentation?

We appreciate that the cross-sectional comparisons of those on the WL with those who have received a transplant may not be novel, but the novelty lies in the fact that we are using genuine measures of QoL as well as controlling for confounding variables and reaching quite different conclusions from those reached when health-status measures alone are used and confounders are not controlled for. We acknowledge that we are limited in the clinical data that have been collected, but that does not mean that assessing PROMs in these two groups is not meaningful.

6. The study design is very difficult to understand still. We are sorry to hear that Reviewer 3 found it difficult to understand, and clarify the points below. We have also added a CONSORT diagram to show participant numbers and measures completed at each stage.

How is this quasi randomized? It was quasi-randomized as not everyone was invited to take part, but rather 'the first eligible patient for each transplant group seen each month (November 2011 to March 2013) by each nurse' was invited to take part (Methods, pph 2). We have removed the reference to quasi-randomized to avoid the confusion.

What do you mean automatic matching? We acknowledge that the sentence included in the paper is unclear, particularly the reference to automatic matching: 'Prevalent listed patients were selected as matched controls for recruited transplant recipients automatically from the UK Transplant Registry database on a fortnightly basis' (Methods, pph 2). To aid clarity, we have changed it to 'The UK Transplant Registry identified possible matched controls for recruited transplant recipients every two weeks, and members of this list of potential participants were then invited to take part as WL patients.' (Methods, pph 2).

How is 165 WL matched to 104 DD and 94 LD? How can age be so different if that is one of the matching variable? The WL group were matched to the DD recipients only, not the LD recipients, which is why age is different for the LD recipients compared with the WL/DD groups. The matching is referenced in the Methods pph 2, described in more detail in the paper by Oniscu et al 2016 published previously in the BMJ Open describing the ATTOM protocol. We have specified in the manuscript that it was only the DD recipients who were matched (Methods, pph 2).

The WL group were matched to the DDKT group, but the final numbers recruited are different, as some participants had a subsequent transplant during data collection. We therefore recruited more participants who were wait-listed so that the final numbers in each group at 12m post-recruitment would be more comparable to one another. So, the 165 recruited as WL patients at 12m, ended up being 98, as 67 had a subsequent transplant during the data collection period.

What was the matching frame? The matching frame is clearly stated in the manuscript: 'based on 'renal centre, age (+/- 5 years), time on the waiting list (+/-100 days), and previous type of RRT' (Methods, pph 2).

7. The qualitative and quantitative findings don't seem to complement each other organically.

The quantitative study focuses on the full range of PROMs in DDKT, LDKT, and WL groups, with some of the sample followed up from pre-transplant to post-transplant. The qualitative study focuses specifically on QoL in those who have had a transplant, as the objective for the qualitative study was to gather a more in-depth understanding of transplant recipients' experiences and how their QoL is impacted by transplantation. We feel that the findings from the qualitative and quantitative aspects of the study do complement one another, and have added the following to the Discussion (pph 2): 'The qualitative interviews show that although participants report improvements to their QoL through a return to 'normal' and improvements in lifestyle, participants still experience ongoing negative impact. Although transplantation is considered a positive and successful treatment, it cannot minimise the negative impact of a renal condition completely.'

We have added the following to the introduction of the manuscript: 'The addition of qualitative research methods alongside PROMs, can be valuable in providing more detailed insight into how transplant recipients experience transplantation and how it impacts QoL.' (Introduction, paph 4).

8. Apparently participants completed treatment QoL and satisfaction 3 mo post tx.

The quality of life and treatment satisfaction measures were completed three months post-transplant once patients had had an opportunity to experience their transplant and associated medication and something of the impact on their lives. Authors are concerned that recipients are not stable enough at 1 year post transplant. How was this 3 month timeframe selected then?

The RDQoL (QoL measure) asks participants to rate how good or bad they feel their life to be, and the RTSQ status treatment satisfaction measure asks participants to think about their treatment in the past few weeks. In contrast, the health-status measure (EQ-5D) asks participants to rate their health today. For these reasons, QoL and treatment satisfaction were completed at three-months to allow participants to experience their new treatment (i.e. transplant) long enough to be able to reflect on how their QoL and treatment satisfaction were impacted after their return home following surgery. Three months was the earliest sensible time point for both the RTSQ and RDQoL to be given and had nothing to do with reaching a stable state. We have added in this explanation in the Methods section, Study design, participants and procedures pph 2.

It was of course only after we had collected data at 12 months that it became apparent that many respondents had yet to return to work and may perhaps go on to have further improvements in their health and quality of life outcomes.

Also, in a response to previous questions author state: If patients received a transplant during the follow-up period, they were asked to complete the measures 12 months post-transplant; they were not asked to complete the measures shortly after transplant, as those recruited. I see a bit of contradiction here.

There are two main reasons why those who received a transplant during the follow-up period were not asked to complete the measures 3m post-transplant. The first reason was ethical. We specified a maximum number of times we would ask respondents to complete questionnaires both to the ethics committee who approved the study and to the patients who consented to take part. We could not ethically exceed this maximum. We adhered to a limit of collecting data at four points only for each participant, with the standard procedure including at least three data collection points: (1) recruitment, (2) 3 months (3) 12m follow-up, and, for some, (4) qualitative interview. If a person recruited as a WL patient was contacted at 12m post-recruitment and informed the researchers that they had had a transplant during this time, we were able to postpone collecting data until they were at 12m post-transplant, whilst still allowing for the option of inviting them to take part in the qualitative interviews. If we had included a second set of questionnaires at 3m post-transplant, then we would have been unable to invite these participants for interview. A second, logistical issue is that when contacting participants at 12m post-recruitment, it was not always possible to contact people within three months of receiving their transplant. We were, however, able to contact everyone who had a subsequent transplant at 12m post-transplant. None of this information was included in the revision to limit the increase in word count, however, the editor has agreed that we can exceed the word count of our manuscript. This information has now been added to the Methods, Study design, participants and procedures section, pph 2.

9. Arguments about why outcomes assessed at 1 yr post - transplant is not convincing.

Authors state: One-year follow-up was selected as part of the ATTOM programme design; and reflects the fact that in this research, follow-up from transplant is commonly measured at one-year post-transplant. This does not sound like a strong rationale.

We are sorry that the reviewer feels we have not justified our rationale; we have included the following in the manuscript: 'One-year follow-up was chosen because it would be expected that clinical outcomes would be stable one-year post-transplant, but it's not clear whether non-clinical outcomes (i.e. quality of life and other PROMs) would also be stable. At the same time, choosing one-year follow-up allows for comparisons to be made with previous research that has examined health status using the same timeline' (Methods, Study design, participants and procedures section, pph 2).

At the same time: Authors mention in discussion " One-year follow-up may be too soon to show differences across transplant groups. ". Why was 1 yr selected then?

As stated above, choosing one-year follow-up means that we could compare our results to previous research that has examined health status using the same timeline. In the first revision, we removed the sentence 'One-year follow-up may be too soon to show differences across transplant groups' from the manuscript, as we acknowledged the potential for confusion. It was instead changed to: 'Most patients had not returned to work at follow-up; those who had returned to work had varying levels of QoL compared to those who had yet to return to work. The long-term impact of taking immunosuppressants, including the increased risk of cancer and other diseases, may not yet be reflected in participants' QoL and other PROMs within the first year of having a transplant. Longer follow-up, therefore, may be required to show any differences across transplant groups, and the long-term impact of immunosuppressant medication on QoL' (Discussion, pph 4). It was only during the qualitative interviews that it became apparent that many transplant recipients had not returned to work by 12 months. We were not aware of this before the study began and it did not therefore influence the selection of a 12-month time point for follow-up. As stated above in point 9, we have included further rationale for why we chose a one-year follow-up.

Further, in their response authors state: Most patients had not returned to work at follow-up, and the effect of immunosuppressant medication on QoL and other PROMs may not be apparent until after one-year post-transplant. KT recipients stabilize clinically at about 3-6-month post-transplant. At 1 yr, several of the measured characteristics are stable.

It may well be that clinical outcomes stabilise at 3-6 months post-transplant, but because no other studies have used genuine measures of QoL, it was unknown how long it would take for QoL and other PROMs to stabilise.

Longer follow-up would be useful to determine how QoL and other PROMs change beyond 12 months. For example, we noted that those who have yet to return to work had worse QoL than those who have returned to work. Additionally, the long-term impact of taking immunosuppressants, including the increased risk of cancer and other diseases, may not yet be reflected in participants' QoL and other PROMs within the first year of having a transplant. This has been added to the Discussion (pph 4).

10. Co-variable selection should be done on theoretical basis not based on association in the given sample.

Choosing covariates based on an underlying theory is an important first step in conducting ANCOVA, but our write-up in the manuscript did not explain this, due to word count constraints. We have included more discussion of the theoretical rationale for the inclusion of the covariates: Clinical factors such as previous renal replacement therapy, and demographic factors including age, and indicators of SES (such as car ownership and qualifications), can influence a person's perspective on their QoL and treatment satisfaction. We therefore assessed these variables; and controlled for them in the analyses if differences between groups were identified on those variables (using Chi Squared tests or ANOVAs).³⁶ (Methods – Analysis section, pph 1).

11. one of the hypotheses mentioned states: It was hypothesized that patients still on the WL would have more negative scores on PROMs than those who received a transplant Does this hypothesis still need an analysis? Is this a question?

We have tested the hypothesis that patients still on the WL will have worse scores on PROMs than those who received a transplant and reported this in the Results, Quantitative findings, pph 2. Are you suggesting we did not need to test this? Previous work has not included the range of PROMs that were included in ATTOM and we were in a position to determine for the first time if our genuine measures of QoL and treatment satisfaction show significant benefits without also showing disadvantages of kidney transplant. We acknowledge that it may be intuitive that those with a transplant would report better scores than those still waiting for a transplant. However, there is a danger in assuming that transplantation is always better, without assessing it adequately.

12. It appears that the sample practically did not include South Asian participants. How did this happen?

For the larger ATTOM programme we undertook the linguistic validation of the EQ-5D and W-BQ12 into multiple South Asian languages and Polish where this work had not already been done using gold standard procedures. These language versions were used at recruitment and one follow-up in the wider ATTOM programme. However, the cost of such linguistic validation work for all the PROMs used in the detailed PROMs sub-study exceeded the resources available to the programme. We therefore limited our detailed PROMs sample to those who were fluent in English. This had an impact on the ethnic breakdown of our sample, which meant that it did not reflect the overall population (approx. 9.4% of transplant recipients recruited to the larger ATTOM research programme were of Asian ethnicity (with the vast majority of South Asian descent)). We acknowledged this in the Discussion (the sample is less ethnically diverse than the UK renal population as a whole), but we have made this more explicit in the Discussion, pph 4.

13. Authors state that qualitative-study participants were representative of the larger samples - how was that assessed and achieved? Is that necessary for qualitative research?

It is not a requirement for qualitative studies to be 'representative' of a population, but in this context, we referred to it as 'representative' because it reflected the age and sex breakdown of the overall sample. To be more precise, we have rephrased to: 'the profile of the qualitative-study participants reflected the age and sex profile of the larger sample'. (Methods - Study design, participants and procedures section, pph 3)

14. In discussion authors state: However, once baseline differences in factors such as age, education, and time on dialysis are controlled for in the subset of patients with true baseline measures pretransplant, DD and LD kidney transplant recipients are found to have similar outcomes post-transplant. In a response to previous question they stated: Unfortunately, we do not have information about how long participants were on dialysis for.

Apologies for what may appear to be a contradiction: it is not in fact a contradiction but needs clarification. We do not have information about how long participants who were recruited as incident transplant recipients were on dialysis (104 DD, 94 LD recipients) prior to recruitment to ATTOM. We do, however, have these data for those originally recruited to ATTOM as wait-listed patients (N=145), who then subsequently had a DD (n=41) or LD transplant (n=26), or remained wait-listed (n=98). The sentence in the Discussion refers to these participants: for them we were able to control for time on dialysis. This clarification has been added to the manuscript (Discussion, pph 1).

15. To a question about adjusting for additional variables authors state: "We agree that including analyses using clinical outcomes would be useful, but the focus of this study was on PROMs and we do not have access to data regarding delayed graft function."

If the research question is PROM/QoL 1-year post-transplant, when designing data collection one would likely want to collect data likely associated with those outcomes. Post-transplant events (length of hospital stay, complications, rejection, DGF are such variables) since they have been repeatedly shown to be associated with additional post-transplant outcomes.

None of the patients included in the present manuscript experienced graft rejection and that is explicit in the manuscript (no one reported a failed transplant during data collection, limiting the scope of the findings to those who remained healthy enough to be on the WL, and those with a functioning graft; Discussion, pph 4). We have acknowledged the limitation that we did not have clinical follow-up data at 12 months. What we have are follow-up data at one-year post-transplant, collected from the participants directly including self-reported health status as well as QoL and the impact of the renal condition on QoL. We can reasonably expect that if complications of transplant and delayed graph failure were greater in the DD recipients than in LD recipients with consequences for other post-transplant outcomes, we would see those differences reflected in worse self-reported health status by DD recipients, and perhaps QoL, but this was not the case. Despite the limitations, the findings are novel and have important clinical implications and we trust that their publication will facilitate the slow paradigm shift away from the use of health-status tools as if they were measuring QoL to the use of genuine measures of QoL and other PROMs that are at least as important to patients as is their health.